# Probabilistic retrieval of volcanic $SO_2$ layer height and partial column density using the Cross-track Infrared Sounder (CrIS)

David M. Hyman[1] and Michael J. Pavolonis[1,2]

[1]Cooperative Institute for Meteorological Satellite Studies (CIMSS), University of Wisconsin - Madison, WI
[2]National Oceanic and Atmospheric Administration (NOAA) Center for Satellite Applications and Research (STAR), Madison, WI

**Correspondence:** David M. Hyman (dave.hyman@ssec.wisc.edu)

**Abstract.** During most volcanic eruptions and many periods of volcanic unrest, detectable quantities of sulfur dioxide ($SO_2$) are injected into the atmosphere at a wide range of altitudes, from ground level to the lower stratosphere. Because the fine ash fraction of a volcanic plume is, at times, collocated with $SO_2$ emissions, global tracking of volcanic $SO_2$ is useful in tracking the hazard long after ash detection becomes dominated by noise. Typically, retrievals of $SO_2$ vertical column density (VCD) have relied heavily on hyperspectral ultraviolet measurements. More recently, infrared sounders have provided additional VCD measurements and estimates of the $SO_2$ layer altitude, adding significant value to real-time monitoring of volcanic emissions as well as climatological analyses. These methods can provide fast and accurate physics-based retrievals of VCD and altitude without regard to solar irradiance, meaning that they are effective day and night and can observe high latitude $SO_2$ even in the winter.

In this study, we detail a probabilistic enhancement of an infrared $SO_2$ retrieval method, based on a modified trace-gas retrieval, to estimate $SO_2$ VCD and altitude probabilistically using the Cross-track Infrared Sounder (CrIS) on the Joint Polar Satellite System (JPSS) series of satellites. The methodology requires the characterization of real $SO_2$-free spectra aggregated seasonally and spatially. The probabilistic approach replaces altitude and VCD estimates with probability density functions for the layer height and the partial VCD at multiple heights, fully quantifying the retrieval uncertainty and allowing the estimation of $SO_2$ partitioning by layer. This framework adds significant value over basic VCD and altitude retrieval because it can be used to assign probabilities of $SO_2$ occurrence to different atmospheric intervals.

We highlight analyses of several recent significant eruptions including the 22 June, 2019 eruption of Raikoke volcano, Kuril Islands; the mid-December, 2016 eruption of Bogoslof volcano; and the 26 June, 2018 eruption of Sierra Negra volcano, Galapagos Islands. This retrieval method is currently being implemented in the VOLcanic Cloud Analysis Toolkit (VOLCAT), where it will be used to generate additional cloud object properties for real-time detection, probabilistic characterization, and tracking of volcanic clouds in support of aviation safety.

## 1 Introduction

During most volcanic eruptions and many periods of volcanic unrest, detectable quantities of sulfur dioxide ($SO_2$) are injected into the atmosphere at a wide range of altitudes, from ground level to the lower stratosphere. Often early in eruptions the fine ash

fraction of a volcanic plume is collocated with $SO_2$ emissions and so ash tracking can be performed by proxy; however, later ash and $SO_2$ tend to evolve along different trajectories due to subtly differences in altitude and removal processes (Karagulian et al., 2010; Corradini et al., 2010; Sears et al., 2013; Moxnes et al., 2014). Early collocation of $SO_2$ and ash is highly significant for informing forward trajectory models (e.g., HYSPLIT) of volcanic clouds as is performed in response to Volcanic Ash Advisories (VAAs) reported by the global network of Volcanic Ash Advisory Centers (VAACs). Because fine ash and $SO_2$ eventually diverge along different trajectories, due in large part to wind shear, layer height estimates are critical for ash and $SO_2$ cloud estimates. Although volcanic ash presents a demonstrated threat to aviation (e.g., ICAO, 2012; Casadevall, 1994; Prata and Rose, 2015; Guffanti et al., 2010), $SO_2$ also presents an aviation safety concern, mainly as a human health hazard and damage by sulfuric acid, as well as impacts on global climate and air quality (Chin and Jacob, 1996; Prata, 2009; Carn et al., 2009; Robock, 2000).

Globally, measurements of $SO_2$ vertical column density (VCD, here in Dobson Units (DU), 1 DU = 2.69 $2.69 \times 10^{16}$ molecules cm$^{-2}$) have previously relied heavily on low-earth orbiting hyperspectral ultraviolet (UV) instruments including the Ozone Monitoring Instrument (OMI), Ozone Mapping and Profiler Suite (OMPS), and Global Ozone Monitoring Experiment–2 (GOME-2) (e.g. Krotkov et al., 2010; Carn et al., 2017; Li et al., 2017; Theys et al., 2013). More recently, efforts to improve UV methods have focused on high-cadence UV measurements made from the Deep Space Climate Observatory - Earth Polychromatic Imaging Camera (DSCOVR-EPIC) at the Earth-Sun (L1) Lagrange point (Carn et al., 2018) as well as high spatial resolution $SO_2$ VCD and limited layer height measurements from the Tropospheric Monitoring Instrument (TROPOMI) (Theys et al., 2019; Hedelt et al., 2019). In the last decade, infrared sounders such as the Infrared Atmospheric Sounding Interferometer (IASI) have provided additional $SO_2$ VCD measurements and estimates of the layer altitude, providing significant added value to real-time monitoring of volcanic emissions as well as climatological analyses (Walker et al., 2011, 2012; Carboni et al., 2012, 2016; Clarisse et al., 2014; Bauduin et al., 2016). These methods can provide fast and accurate physics-based retrievals of VCD and altitude. Furthermore, because these techniques principally rely on thermal contrast in the atmosphere and not solar irradiance (as in UV measurments), they are effective day and night and can observe high latitude $SO_2$ in the winter months when UV techniques are unavailable. This twice-daily global coverage makes IR-based $SO_2$ retrieval a highly useful tool for operational support of aviation safety as well as truly continuous global analysis of $SO_2$ from volcanic eruptions.

In this study, we detail a probabilistic enhancement of the infrared $SO_2$ retrieval method of Clarisse et al. (2014), based on a modified trace-gas retrieval (Walker et al., 2011) to estimate $SO_2$ VCD and altitude probabilistically utilizing the Cross-track Infrared Sounder (CrIS) currently aboard the Suomi-NPP (SNPP) and NOAA-20 satellites as part of the Joint Polar Satellite System (JPSS), having a local time ascending node (LTAN) of 1:30 PM with NOAA-20 operating approximately 50 minutes ahead of SNPP. Similar to IASI, CrIS is a Fourier transform Michelson interferometer covering three regions of the infrared spectrum: long-wave infrared (LWIR) (650-1095 cm$^{-1}$), mid-wave IR (MWIR) (1210-1750 cm$^{-1}$), and short-wave IR (SWIR) (2155-2550 cm$^{-1}$) (Han et al., 2013). Of the two principal $SO_2$ absorption features in the infrared ($\nu_1$: 1000-1200 cm$^{-1}$, $\nu_3$: 1300-1410 cm$^{-1}$), only the $\nu_3$ band is covered by CrIS (MWIR). As highlighted by Carboni et al. (2012) and Clarisse et al. (2014), $\nu_3$ is the stronger of the two absorption bands; however, it does contain significant interference from water vapor,

limiting this retrieval's ability to characterize $SO_2$ features at very low altitudes. However, it is exactly the variable amounts of interference with water vapor at different heights that gives this technique its ability to retrieve $SO_2$ altitude information (Clarisse et al., 2014). As these studies pointed out, although clouds poses similar absorption features to water vapor, it is only in a broadband sense, allowing the finer absorption lines of $SO_2$ to be distinguished even scenes with overlying meteorological clouds as long as the clouds are not nearly opaque. Although the interference from water vapor is a significant theoretical limitation of this approach, in practice, we have been able to extract some information on low-altitude $SO_2$ clouds even in the tropics which is detailed later. Despite these limitations, the $\nu_3$ band absorption lines are only minimally influenced by ash and dust, making this height retrieval method especially useful early in the evolution of volcanic eruption clouds where there is typically collocation between $SO_2$ and ash clouds.

Both CrIS instruments are currently operating in full spectral resolution mode (FSR), providing MWIR spectra at 0.625 cm$^{-1}$ spectral resolution since December 2015 for SNPP CrIS (excepting a major outage 26 March - 1 August, 2019) and February 2018 for NOAA-20 CrIS. CrIS scans consist of 30 fields of regard (FOR) in 3.3° steps between ±48.3° scan angle, each of which contains 9 circular fields of view (FOV) arranged in a square ($3 \times 3$) array which rotates and stretches as the mirror moves away from nadir towards edge of scan (Han et al., 2013). CrIS granules are collected into 6-minute granules of 45 scans, resulting in 12,150 MWIR FSR spectra collected every 6 minutes. The CrIS swath width is 2,200 km; however, because of the rotating FORs, some ground points are measured by multiple FOVs even within the same scan and some gaps exist due to the square FOR layout of circular FOVs and the presence of short gaps between scans. The FOV at the center of each FOR (number 5) is a 14 km-diameter circle at nadir, extending out to an 43.6 km $\times$ 23.2 km (major and minor axes) ellipse for the first and last FORs on the edge of the swath (Han et al., 2013; Wang et al., 2013). Although CrIS FOVs are slightly larger than IASI FOVs (14 km vs. 12 km at nadir), there are many more of them per scan since CrIS FOR are $3 \times 3$ arrays whereas IASI FOR are $2 \times 2$ with larger gaps between FOV, FORs, and scan lines but the same swath width (e.g., Sun et al., 2018). Consequently, CrIS makes many more measurements per area than does IASI, resulting in greater overall resolution that IASI. Lastly, all of the CrIS MWIR channels used in the present study are, in general, very low noise (noise equivalent differential radiance, NEdN$< 0.05$ mW m$^{-2}$ sr$^{-1}$ cm) with the exception of SNPP CrIS FOV 7, which is above specification. Later in this study we will show some retrievals from this FOV; however, these are considered as being of very low quality and are not considered reliable. They are shown here only to elucidate how strong instrument noise is propagated to the retrieval.

The NOAA Unique CrIS/ATMS Processing System (NUCAPS) already includes a retrieval of $SO_2$ from CrIS data (Gambacorta, 2013); however, it is based on a heritage algorithm designed to estimate many trace gases from cloud cleared radiances in one retrieval whereas we focus more specifically on the problem of retrieving $SO_2$ in any background atmosphere from all available CrIS measurements. The methodology requires the characterization of the background mid-wave infrared spectrum of the $SO_2$-free atmosphere, which is done by collecting the statistics of more than 360 million $SO_2$-free CrIS spectra aggregated seasonally and spatially. The probabilistic approach replaces altitude and VCD estimates with a non-parametric probability density function (PDF) for the layer height and estimates (with uncertainty) of the partial VCD at multiple heights, fully quantifying the retrieval uncertainty and allowing the estimation of $SO_2$ partitioning by layer (Fig. 1). This framework adds significant value because it can be used to assign probabilities of $SO_2$ occurrence in different intervals of the atmosphere,

which could prove very useful for aviation safety in the future when changing aviation hazard priorities will require such information (ICAO/IAVW, 2019).

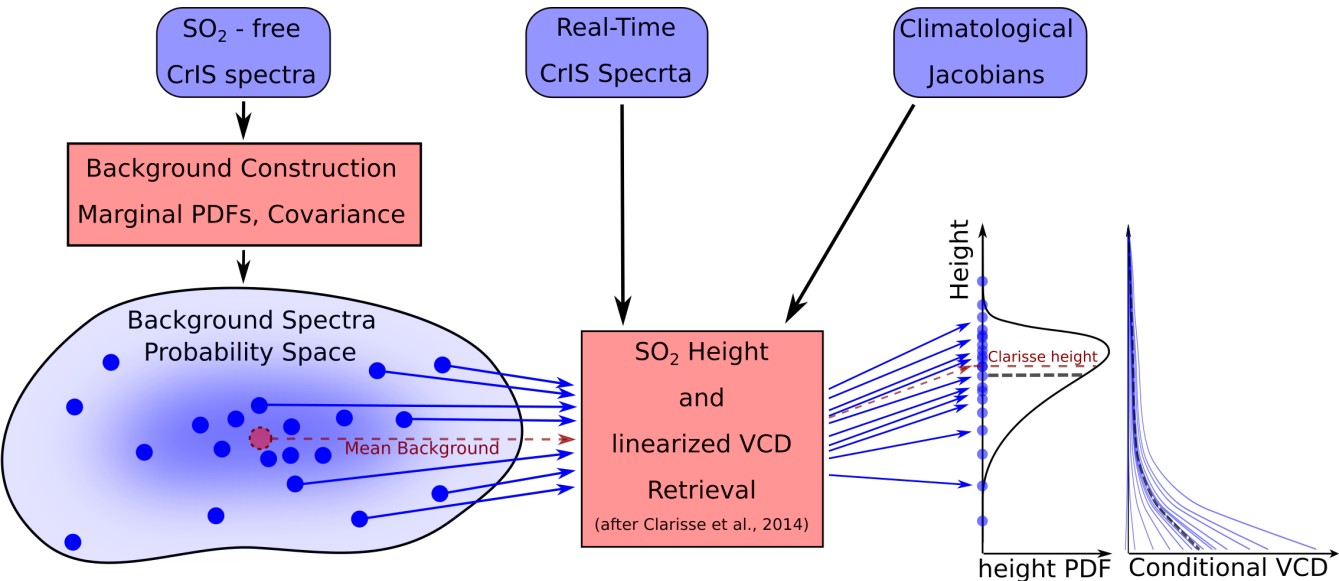

**Figure 1.** : *Flowchart showing probabilistic framework for Monte Carlo height and VCD, yielding a PDF for the height which is not generally Gaussian and may be heavily skew and a Gaussian distribution of conditional VCD ($\hat{X}|H=h$). The height retrieval of Clarisse et al. (2014) is shown schematically in red lines, giving a single height estimate which is not in general the mean height (approximately the black dotted line in the height PDF). As this figure is only schematic, the pictorial relationship between these height estimates in not universal.*

In this study we analyze several recent significant eruptions including the 22 June, 2019 eruption of Raikoke volcano, Kuril Islands; the mid-December, 2016 eruption of Bogoslof volcano; and the 26 June, 2018 eruption of Sierra Negra volcano, Galapagos Islands. This retrieval method is currently being implemented in the VOLcanic Cloud Analysis Toolkit (VOLCAT,
https://volcano.ssec.wisc.edu/ ; Pavolonis et al., 2013, 2015a, b, 2018), where it will be used to generate additional cloud object properties for real-time detection, probabilistic characterization, and tracking of volcanic clouds in support of aviation safety.

## 2  Probabilistic SO$_2$ layer retrieval theory

### 2.1  Classical methods for height retrieval

As a preliminary we discuss several methods which we describe here as "classical". In fact these methods are relatively recent; however, they do not make full use of the probability spaces which we will exploit here. Previous analyses of the height and distribution of volcanic SO$_2$ plumes using data from IASI by Carboni et al. (2012), Clarisse et al. (2014), and Carboni et al. (2016) utilized trace gas methods modified from the method originally outlined by Walker et al. (2011). The analysis of

Carboni et al. (2012) imposed an *a priori* Gaussian vertical distribution over pressure coordinates for the $SO_2$ concentration, retrieving the total $SO_2$ VCD, the mean pressure and the standard deviation pressure (spread, only if VCD sufficiently strong). By contrast, Clarisse et al. (2014) developed a system in which the $SO_2$ is assumed to exist in a narrow box-profile layer and an iterative retrieval is performed for the VCD concentration conditional on the retrieved $SO_2$ layer altitude. The principal differences between these methods and the method detailed here are summarized in Table 1. Employing the notation of Rodgers (2000), this retrieval relates a set of parameters governing the concentration of a trace gas comprising the true state ($\boldsymbol{x} \in \mathbb{R}^M$, $SO_2$ in this case) to a set of measurements $\boldsymbol{y} \in \mathbb{R}^N$ (typically brightness temperature spectra) by a forward radiative transfer model $\boldsymbol{F} : \mathbb{R}^M \to \mathbb{R}^N$. In the following exposition, we focus on and enhance the method of Clarisse et al. (2014).

The infrared trace gas methods of Walker et al. (2011) and Clarisse et al. (2014) rely on the ability to write the data and true state each as a sum of their respective climatological background averages ($\overline{\boldsymbol{y}}_{bg}, \overline{\boldsymbol{x}}_{bg}$) and their anomalies ($\widetilde{\boldsymbol{y}}, \widetilde{\boldsymbol{x}}$). Linearizing around the climatological average gives:

$$\overline{\boldsymbol{y}}_{bg} + \widetilde{\boldsymbol{y}} = \boldsymbol{F}(\overline{\boldsymbol{x}}_{bg}; \boldsymbol{u}) + \mathbf{K}\widetilde{\boldsymbol{x}} + \varepsilon_{tot} \tag{1}$$

where $\boldsymbol{u}$ is a collection of all auxiliary parameters necessary to the forward model including atmospheric pressure, temperature, and water vapor profiles, as well as the state of the surface and instrument. $\mathbf{K} \in \mathbb{R}^{N \times M}$ is the Jacobian of $\boldsymbol{F}$ linearized about $\overline{\boldsymbol{x}}_{bg}$ and $\varepsilon_{tot}$ is the total error associated with the measurement, linearization, and other surface and atmospheric properties (including clouds) that influence the measured radiances. Because it is inferred that $\overline{\boldsymbol{y}}_{bg} = \boldsymbol{F}(\overline{\boldsymbol{x}}_{bg}; \boldsymbol{u})$, the equation for the error is reduced simply to a relationship between the state and measurement anomalies:

$$\varepsilon_{tot} = \widetilde{\boldsymbol{y}} - \mathbf{K}\widetilde{\boldsymbol{x}}. \tag{2}$$

From this formula, the retrieval of $\widetilde{\boldsymbol{x}}$ can proceed either by maximum likelihood estimation, iterative methods such as Levenberg-Marquard and gradient descent algorithms, or other methods.

For a 1 km-thick box-profile layer, the concentration of anomalous $SO_2$ can be represented by two parameters: the total VCD of the gas ($x$) and the height of the layer center ($h$). Using such a profile, the model spectrum is then a function of the VCD and the layer height. In Clarisse et al. (2014) and here, the Jacobian is represented a function of height, but only contains a VCD perturbation. Because this Jacobian only measures the sensitivity of the forward model to the presence of $SO_2$ at each height independently, it is best viewed as a set of vectors in the same space as $\boldsymbol{y}$ rather than as a matrix and thus we write it hereafter as $\boldsymbol{K}(h) \in \mathbb{R}^N$. Additionally, the model Jacobian for the trace gas retrieval is calculated at the background state (zero for $SO_2$), and the Jacobian is approximated as a finite difference at each altitude:

$$\boldsymbol{K}(h) \approx \frac{\boldsymbol{F}(\epsilon, h; \boldsymbol{u}) - \boldsymbol{F}(0, h; \boldsymbol{u})}{\epsilon} \tag{3}$$

where the perturbation ($\epsilon$) is taken as 5 DU.

In the present study, we pre-compute a limited database of Jacobians for 1 km-thick $SO_2$ layers centered at 28 altitudes between 0 and 32 km (Fig. 2b, here; Fig. 1 of Clarisse et al. (2014)). We use standard profiles of pressure, temperature, and water vapour for a tropical atmosphere, summer and winter mid-latitude atmospheres, and summer and winter subarctic

**Table 1.** Summary of Recent Infrared $SO_2$ Height Estimation Methods

| Method: | Carboni et al. (2012) | Clarisse et al. (2014) | This Study |
|---|---|---|---|
| Instrument: | IASI | IASI | CrIS |
| $SO_2$ Band: | $\nu_1 + \nu_3$ (1000–1200 cm$^{-1}$ + 1300–1410 cm$^{-1}$) | $\nu_3$ (1300–1410 cm$^{-1}$) | $\nu_3$ (1300–1410 cm$^{-1}$) |
| Retrieved Quantities: | $SO_2$ center height (pressure), total VCD, spread (pressure) | $SO_2$ layer height, total VCD | $SO_2$ height PDF, partial VCD at each height |
| $SO_2$ Profile Type: | Gaussian (pressure) ($\approx$ Log-normal, altitude) | Box (altitude) | Box (altitude) |
| $SO_2$ Profile Spread: | retrieved (100 mb default) | 1 km | 1 km |
| Height Retrieval: | Joint with VCD | Independent of VCD | Independent of VCD |
| Style: | Levenberg - Marquard | Z-score maximizing | Z-score maximizing |
| VCD Retrieval: | Joint with height | Total (conditional) VCD given retrieved height | Partial VCD weighted by height PDF |
| Style: | Levenberg - Marquard | Levenberg - Marquard | Modified Linear |
| Retrieval Uncertainty: | Posterior covariance matrix (Multivariate Normal) | Height, VCD variances (Independent Bivariate Normal) | Height PDF (nonparametric), partial VCD variances (Normal) |
| Probabilities Retrieved | No | No | Yes |

atmospheres, resulting in a total of 140 Jacobians to be used in the retrieval (Clough et al., 2005). As in Pavolonis (2010), all radiative transfer model simulations used here were performed using the LBLDIS tool (Turner, 2005), which utilizes the Line-by-Line Radiative Transfer Model (LBLRTM; Clough and Iacono, 1995; Clough et al., 2005) to compute gaseous absorption and the Discrete Ordinate Radiative Transfer (DISORT) model to complete the radiative transfer calculation (including multiple scattering).

Following Clarisse et al. (2014), a height-dependent Jacobian can be used to calculate a statistical z-score, measuring the relative confidence in the presence of $SO_2$ at each height:

$$z(h; \boldsymbol{y}, \overline{\boldsymbol{y}}_{bg}) = [\boldsymbol{K}^{\intercal}(h)\mathbf{S}^{-1}\boldsymbol{K}(h)]^{-\frac{1}{2}} \boldsymbol{K}^{\intercal}(h)\mathbf{S}^{-1}(\boldsymbol{y} - \overline{\boldsymbol{y}}_{bg}) \qquad (4)$$

where the mean climatological background spectrum $\overline{\boldsymbol{y}}_{bg}$ and error covariance matrix $\mathbf{S}$ are built from spectral residuals from a database of measurements with low or no detectable $SO_2$ present. The construction of this database of $SO_2$-free spectra is detailed in section 2.5. Note that because $\boldsymbol{K}(h)$ is a vector, the factor $\boldsymbol{K}^{\intercal}(h)\mathbf{S}^{-1}\boldsymbol{K}(h)$ is a scalar at each height $h$. Here, $z(h;\boldsymbol{y},\overline{\boldsymbol{y}}_{bg})$ is the statistical z-score (number of standard deviations from the mean) of finding the $SO_2$ anomaly at altitude $h$ given the data $\boldsymbol{y}$ and the $SO_2$-free background spectrum $\overline{\boldsymbol{y}}_{bg}$. Using the z-score, Clarisse et al. (2014) estimated the layer height (which we refer to as $h_C$) as that which maximizes the z-score function:

$$h_C := \underset{h}{\arg\max}\, z(h;\boldsymbol{y},\overline{\boldsymbol{y}}_{bg}). \tag{5}$$

This method was used in that study to produce a consistent and reasonably accurate set of cloud top height estimates for the 2011 eruption of Nabro Volcano, Eritrea. This method is currently the principal operational $SO_2$ layer height method for IASI used by the Support to Aviation Control Service (SACS, https://sacs.aeronomie.be/) of the Royal Belgian Institute for Space Aeronomy (BIRA-IASB) supporting several Volcanic Ash Advisory Centers (VAACs) in near-real time (Brenot et al., 2014).

For simplicity, throughout the remainder of this work we refer to this type of height retrieval with a function notation:

$$h_C = g(\boldsymbol{y},\overline{\boldsymbol{y}}_{bg}) := \underset{h}{\arg\max}\, z(h;\boldsymbol{y},\overline{\boldsymbol{y}}_{bg}). \tag{6}$$

## 2.2 Probabilistic enhancement

Throughout, we make a distinction between our method as being probabilistic whereas other methods are deterministic; however, we note here that the classical methods are all based on the optimal estimation (a Bayesian method) of Rodgers (2000) and therefore are probabilistic in the sense that the retrieved quantities are the (Gaussian) mean and covariance of the state estimate. We make this distinction to highlight the fact that in the present study we focus on a much more detailed uncertainty propagation, specifically, propagation uncertainty about the $SO_2$-free background atmosphere to the height retrieval, allowing some departure from the Gaussian assumption underlying previous methods. Although the Gaussian assumption is workable for many types of retrieval, it is unsuitable for the Clarisse et al. (2014) height retrieval in particular due to the role played by the $\arg\max$ operation. To see this directly, we must "probabilize" the Clarisse et al. (2014) height retrieval as follows. In this process, we use a notation common in probability theory in which random variables are represented as capitalized versions of their deterministic realizations. In what follows, the only exception to this notation will be that the forward model, Jacobian, and covariance matrix are not random variables.

Instead of calculating the mean and covariance of the climatological background as in a traditional trace gas retrieval, we treat the background ($SO_2$-free) spectrum as a random vector $\boldsymbol{Y}_{bg}$ (rather than the realization $\overline{\boldsymbol{y}}_{bg}$, which is its mean), where the vector elements (each of the sampled wavenumbers or channels) are random variables. Although the total uncertainty on the measured $SO_2$-free background spectrum contains a mix of aleatoric (stochastic in fact) and epistemic (knowledge-deficit) uncertainties, the epistemic uncertainty due to a lack of knowledge about the true $SO_2$-free background for a given spectrum is considerably larger than aleatoric sources such as instrument noise and scattering. Consequently, we generate a probability space for the so2-free spectrum $\boldsymbol{Y}_{bg}$, the uncertainty for which is derived principally from our ignorance of the true atmospheric state if $SO_2$ were not present. The brightness temperatures for each channel (element of $\boldsymbol{Y}_{bg}$) is characterized by

its own marginal probability distribution. In reality, these distributions may belong to a family of parameterized distributions; however, in this study they are non-parametric, only characterized by a marginal distribution (histogram) on each channel. Additionally, the elements of $\boldsymbol{Y}_{bg}$ are correlated which is given by the the covariance structure:

$$\mathbf{S} = \mathbb{E}\Big[(\boldsymbol{Y}_{bg} - \mathbb{E}[\boldsymbol{Y}_{bg}])(\boldsymbol{Y} - \mathbb{E}[\boldsymbol{Y}_{bg}])^{\intercal}\Big] \tag{7}$$

where the expectation is an average over all $SO_2$-free spectra in the database (detailed in Section 2.5).

In this framework, the z-score function is a conditional random variable given the layer height $h$:

$$Z(h; \boldsymbol{y}, \boldsymbol{Y}) = Z|H{=}h = [\boldsymbol{K}^{\intercal}(h)\mathbf{S}^{-1}\boldsymbol{K}(h)]^{-\frac{1}{2}}\boldsymbol{K}^{\intercal}(h)\mathbf{S}^{-1}(\boldsymbol{y} - \boldsymbol{Y}_{bg}) \tag{8}$$

and the height is therefore a random variable $H := g(\boldsymbol{y}, \boldsymbol{Y}_{bg})$.

Implicitly, Clarisse et al. (2014) assumes that $\boldsymbol{Y}_{bg}$ is a multivariate normal random vector with mean $\overline{\boldsymbol{y}}_{bg} = \mathbb{E}[\boldsymbol{Y}_{bg}]$ and covariance $\mathbf{S}$, meaning that $Z$ is a standard normal random variable. This fact about the z-score is expected to hold in the present case with the full probabilistic characterization of the generally non-Gaussian $\boldsymbol{Y}_{bg}$ because the z-score is a weighted sum over all of the channels in $\boldsymbol{Y}_{bg}$, which is expected to converge to a Gaussian for a large collection of channels. Because the function $g$ uses the $\arg\max$ operation, which in not exactly a proper function (and also not linear), we can write that

$$\mathbb{E}[g(\boldsymbol{y}, \boldsymbol{Y}_{bg})] \neq g(\boldsymbol{y}, \mathbb{E}[\boldsymbol{Y}_{bg}]). \tag{9}$$

That is, the random variable resulting from a nonlinear transformation of a Gaussian random variable is not itself Gaussian and the mean value of that new variable is not equal to the value obtained by transforming the mean of the Gaussian (thus, $\mathbb{E}[H] \neq h_C$, Fig. 1 schematically), a standard result in elementary probability theory texts (e.g., DeGroot and Schervish, 2012). Similarly, $h_C$ is not generally the maximum likelihood height either ($h_C \neq \mathrm{mode}[H]$). Consequently, without a clear understanding of what $h_C$ is measuring in terms of the statistics of $H$, it is difficult to contextualize the value $h_C$. The principal enhancement over the classical method comes from setting the height retrieval in a probabilistic framework, enabling precise propagation of uncertainty in the background state to uncertainty in the retrieved height.

This study aims to estimate the probability distribution of $H$ and show the importance of its PDF in making predictions about the cloud. We enhance the method of Clarisse et al. (2014), by retrieving a probabilistic $SO_2$ layer, that is, we retrieve the $SO_2$ layer height as a PDF for the height (Fig. 1). As described, the probabilistic nature of the retrieval product is derived from propagating our uncertainty about the $SO_2$-free background spectrum through the Clarisse et al. (2014) height retrieval method (Fig. 1). Here, we use a set of 10,000 possible $SO_2$-free background spectra computed by Monte-Carlo (MC) sampling according to the collection of marginal distributions of $\boldsymbol{Y}_{bg}$ and its covariance matrix $\mathbf{S}$, which are first computed from a database of $SO_2$-free spectra (detailed in Section 2.5). The process for sampling this generally non-Gaussian correlated random vector is detailed in Appendix A. In this study each sample is denoted $\boldsymbol{y}_{bg}^s \in \Omega_{\boldsymbol{Y}_{bg}}$ where $\Omega_{\boldsymbol{Y}_{bg}}$ is the sample space of $\boldsymbol{Y}_{bg}$.

Although we could directly estimate the height PDF from sampling the many different backgrounds, we treat our retrieval as an update on the Clarisse et al. (2014) height estimate, and cast this process in a Bayesian framework. In this framework, we treat the estimate and uncertainty from the (Clarisse et al., 2014) method as a prior distribution for the height and construct an

approximate likelihood function from the data not accounted for directly in that method (the distribution of the many possible spectral residuals). The height PDF which is sought is the posterior distribution.

We impose a Gaussian prior with mean and variance given by the Clarisse et al. (2014) method. To estimate the mean and variance of the Gaussian, we first retrieve the Clarisse et al. (2014) height $h_C$ and then generate many height estimates around $h_C$ using a model spectral anomaly with $SO_2$ assumed at $h_C$ and MC sampling of the zero-mean noise contained in the collection of possible backgrounds. Specifically, we estimate the height due to noisy model spectral anomaly samples

$$\widetilde{\boldsymbol{y}}^s = \left[ \boldsymbol{F}(\epsilon, h_C, \boldsymbol{u}) - \boldsymbol{F}(0, h_C, \boldsymbol{u}) \right] - \left[ \boldsymbol{y}_{bg}^s - \overline{\boldsymbol{y}}_{bg} \right]. \tag{10}$$

This anomaly represents a modelled spectral anomaly with zero-mean spectral noise added and allows for the possibility of a bias induced by the difference between the mean climatological background and the model background which does not include cloud layers. The Gaussian prior mean and variance are then taken to be the mean and variance of these noisy modelled samples. Of note, this Gaussian prior mean is very close to the value $h_C$, but is preferred since its use does not restrict the Gaussian to be centered only at height values for which Jacobians were computed. Using these values for mean and variance is very similar to the estimate with uncertainty shown in Fig. 2 of Clarisse et al. (2014). This mean and variance parameterize the Gaussian prior distribution $f_H^{prior}(h)$.

The likelihood function is constructed directly by retrieving the height due to the real spectrum $\boldsymbol{y}$ and the set sampled background spectra. Each height sample is generated as:

$$h^s = g(\boldsymbol{y}, \boldsymbol{y}_{bg}^s), \tag{11}$$

that is, we construct the random variable from elements of its sample space $h^s \in \Omega_H$. The likelihood function measures the distribution of the possible real spectral residuals $(\boldsymbol{y} - \boldsymbol{Y}_{bg})$ given an $SO_2$ layer at height $h$. As there is no analytic probability model to describe this, we estimate that the likelihood is proportional to the distribution of possible heights as computed by kernel density estimation (KDE, e.g., Silverman, 1986) on this set of height samples:

$$\mathcal{L}(h; \boldsymbol{y} - \boldsymbol{Y}_{bg}) := \hat{f}(\boldsymbol{y} - \boldsymbol{Y}_{bg} \mid h) \propto \mathrm{KDE}(\{h^s\}). \tag{12}$$

An alternative approach would be to replace the KDE distribution by a simple histogram of the samples, although we use KDE because we seek a distribution which is at least piecewise-continuous, not piecewise-constant. This gives an estimate of the posterior height PDF:

$$f_H(h) \propto \mathcal{L}(h; \boldsymbol{y} - \boldsymbol{Y}_{bg}) \, f_H^{prior}(h) \tag{13}$$

where the proportionality is eliminated by normalizing the posterior PDF such that the total probability is unity.

Although slower than retrieving the layer height ($h_C$) due to a mean spectrum alone, this distribution provides significantly more information including the full PDF of $H$. This PDF may be used to calculate the modal, mean, and median values of the retrieved height or probabilities of finding the plume in a given altitude interval. Additionally, this PDF is essential for calculating the VCD correctly according to probability theory as detailed in the following section.

## 2.3 Probabilistic Vertical Column Density

Although this method is used primarily for detection (using z-scores) and height estimation, we estimate VCD as a side-product, which we treat here as a random variable $\hat{X}$. Because we will use a linearized retrieval, the VCD values presented here will not be as accurate as those produced by an iterative technique. However, the details of our process below produce VCD values that are reasonably accurate for all but very strong emissions as is demonstrated later in this work. Specifically, the way in which the height PDF is incorporated (detailed below) mitigates some underestimation error that would otherwise occur in a linearized approach for even dilute $SO_2$ clouds. As with the height estimation, the uncertainty propagated to the VCD primarily represents uncertainty about the $SO_2$-free background, which is of great importance in these linearized trace gas methods.

Because the estimated VCD depends strongly on the layer height, we refer to an estimate of total VCD where the layer is given as a specified height as a "conditional VCD." In this framework, the VCD estimates of Walker et al. (2011, 2012) are conditional VCDs and are represented as a conditional random variable:

$$\hat{X}|H{=}h \; = \; \cos\theta \,[\boldsymbol{K}^{\mathsf{T}}(h)\mathbf{S}^{-1}\boldsymbol{K}(h)]^{-1}\boldsymbol{K}^{\mathsf{T}}(h)\mathbf{S}^{-1}(\boldsymbol{y}-\boldsymbol{Y}_{bg}) \tag{14}$$

where an air mass factor equal to the cosine of the satellite zenith angle ($\cos\theta$) has been applied. This formula is in some sense a probabilistic enhancement of an optimal unconstrained least-squares estimate. Similar to the z-scores, this function is normally distributed at every height with mean $\mathbb{E}[\hat{X} \mid H{=}h]$ and variance $\mathrm{Var}[\hat{X} \mid H{=}h]$. These are calculated as the sample mean and variance of the conditional VCD samples due to the many possible background spectra used to estimate the height PDF. The conditional VCD is a random function of height which generally reflects the principles of the Beer-Lambert Law for any given realization, that is, a smaller VCD is retrieved for a given spectral anomaly if the layer is assumed to be higher in the atmosphere. If the height of the $SO_2$ layer were known exactly, the VCD could be estimated by evaluating the conditional VCD function at that exact height. This is exactly what is done in the VCD retrieval of Clarisse et al. (2014), except using an iterative conditional VCD calculation. However, in this study, since the height of the layer is known only probabilistically, that is as measured by the PDF $f_H(h)$, additional computation is required to determine the VCD.

Although we do not know the true vertical profile of $SO_2$ concentration, the retrieval assumes a thin layer representation of the $SO_2$. The total VCD ($\hat{X}$, a random variable) is obtained by integrating the box profile between the ground and the top of the atmosphere. Similarly, a partial VCD, denoted here as $\hat{X}(h)$, can be calculated by integrating the box profile between the ground and some height $h$ which is zero for $h < H$ and rises linearly within the layer to $\hat{X}$ for $h \geq H$. Because the assumed concentration profile scales linearly with the total VCD ($\hat{X}$) and the conditional VCD is normally distributed at each height, the partial VCD is normally distributed as well, thus requiring only two parameters: the mean and variance which can be found using the conditional VCD function calculated above.

Here, we give approximation formulae for the mean and variance partial VCD. The derivation of these formulae are detailed in Appendix B. The mean partial VCD below height $h$ is found by the law of total expectation:

$$\mu_{\hat{X}}(h) := \mathbb{E}[\hat{X}(h)] = \int_0^h f_H(\eta)\, \mathbb{E}[\hat{X} \mid H=\eta]\, \mathrm{d}\eta \tag{15}$$

where the expectation of the conditional VCD is taken as the mean of the samples. This formula represents a weighted average of mean conditional VCD values where the height probability density assigns the weights. Because the conditional VCD is a function of height, this formula mixes the VCD estimates from different assumed heights.

The variance can also be calculated from the statistics of the conditional VCD expectation:

$$\sigma_{\hat{X}}^2(h) := \mathrm{Var}[\hat{X}(h)] = \int_0^h f_H(\eta)\left[\mathrm{Var}[\hat{X} \mid H=\eta] + \left(\mathbb{E}[\hat{X} \mid H=\eta]\right)^2\right] \mathrm{d}\eta - \mu_{\hat{X}}^2(h) \tag{16}$$

where the conditional mean and conditional variance were previously estimated from the MC samples.

The covariance between the partial VCDs of two altitudes ($a$ and $b$) is given in Appendix B. Of particular interest, these formulae may be used to calculate the expectation and variance values of the partial VCD between two altitudes:

$$\begin{cases} \mathbb{E}[\hat{X}(b) - \hat{X}(a)] = \mu_{\hat{X}}(b) - \mu_{\hat{X}}(a) & \text{(17a)} \\ \mathrm{Var}[\hat{X}(b) - \hat{X}(a)] = \sigma_{\hat{X}}^2(b) + \mu_{\hat{X}}(a)\left(\mu_{\hat{X}}(b) - \mu_{\hat{X}}(a)\right). & \text{(17b)} \end{cases}$$

In this system, we retrieve probabilistic $SO_2$ information in two stages. In the first stage we perform an initial detection using the classical method (Eq. 4) to pre-screen each CrIS FOV that likely contains $SO_2$, taken as an initial maximum z-score greater than 5, that is, $z(h_C; \boldsymbol{y}, \overline{\boldsymbol{y}}_{bg}) > 5$ (e.g., Walker et al., 2011, 2012; Clarisse et al., 2014). Preliminary investigation of this threshold indicates that it somewhat conservative, striking a balance between including regions of diffuse $SO_2$ and excluding almost all false detections. In the second stage, we retrieve the height PDF and the mean and variance partial VCD as a function of height for each CrIS FOV that satisfies this initial z-score threshold.

## 2.4 Specialized Retrieval for Strong $SO_2$ Loading

For strong $SO_2$ columns, an alternate retrieval is needed to increase sensitivity of the retrieved VCD owing to error induced by the linearized retrieval. We define strong loading here heuristically as $z > 200$ which in preliminary testing corresponded to VCD values between approximately 10 DU and 20 DU. Since the conditional VCD retrieval uses a linearized forward model with only a 5 DU perturbation, it is expected that such an approximation would only hold for values near 5 DU and that many physically realistic VCD values would fall outside the linearization's radius of convergence. For large VCD values, the sensitivity of linear Jacobians to additional $SO_2$ is greatly reduced for most CrIS channels, especially the strongest CrIS channels, so linearized Jacobians with only a 5 DU anomaly will drastically under predict the VCD for a given brightness temperature difference (Fig. 2b). In order to construct a linearized Jacobian that is more sensitive at higher VCD values, two approaches are possible: (i) we could use a larger VCD perturbation (a coarser finite difference) or (ii) we could seek a special

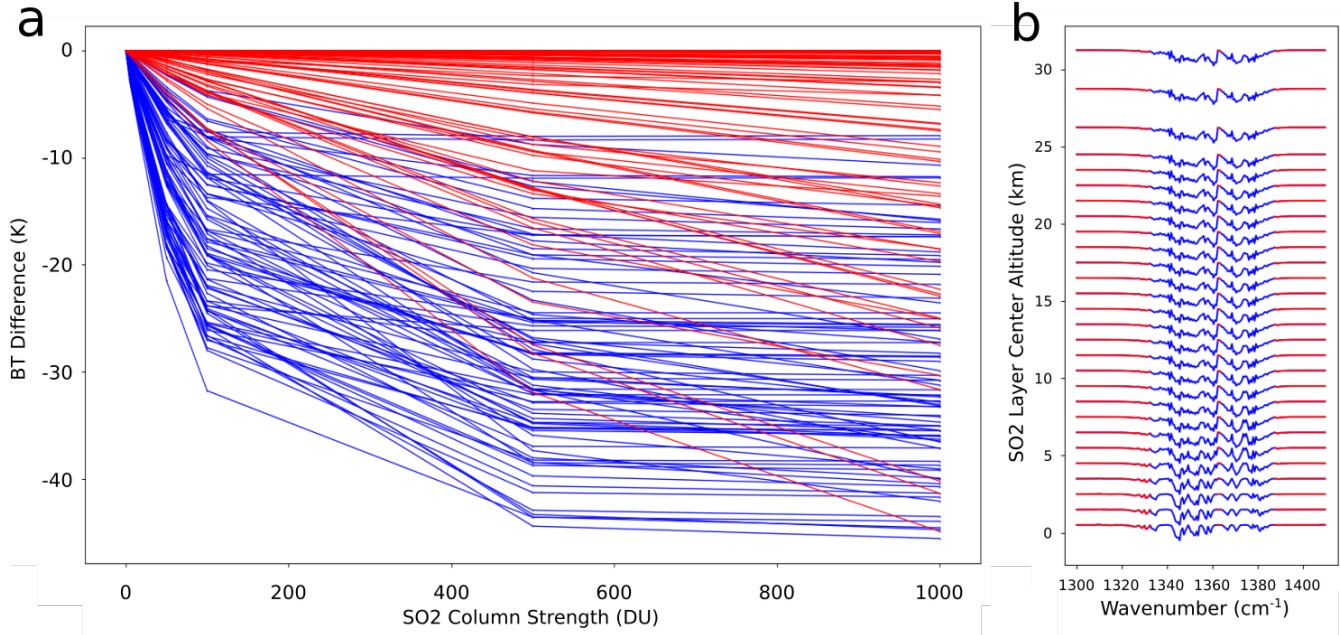

**Figure 2.** : *a) Channel-wise brightness temperature difference response to increasing VCD. All channels here are used in the original retrieval. Red channels are used in the specialized retrieval due to their mostly linear response across the full range of reasonable VCD. b) Height dependent Jacobians (normalized) showing specialized channel selection in (a).*

combination of channels for which the linearization is a good approximation. The first approach is limited in that the strongest responses in the forward model are highly nonlinear so a coarse finite difference will induce large errors there. The second approach is promising as long as such a suitable channel selection can be made which still preserves some of the main features of the $\nu_3$ absorption band, retains enough channels to be robust, and is only applied when it is certain that $SO_2$ is dominating the signal.

In addressing this issue, we adopt the second approach. The specialized Jacobian must be dominated by channels with approximately linear forward model responses (Fig. 2b). This was accomplished in practice by constructing a sequence of Jacobians with various finite difference coarseness and choosing those channels for which the sequence of Jacobians are approximately constant. This channel subset (Appendix D) is used with the original 5 DU Jacobian which can then be extrapolated successfully to high VCD values because the forward model truly is approximately linear for those channels.

One complication here is the fact that the channels with the most linear response are also those which are least sensitive to $SO_2$ VCD (Fig. 2). However, by applying this new retrieval only when strong $SO_2$ loading ($z > 200$) is detected by the pre-screening retrieval, the signal is guaranteed to be dominated by $SO_2$ absorption even in the weak, approximately linear channels and the linearization is expected to have moderately good accuracy. In theory, a sequence of increasingly restricted retrievals could be implemented to increase the sensitivity to even stronger $SO_2$ loads; however, even the most sensitive channels in the specialized subset show the worst linear approximations 2a) and so some underestimation in the inversion will always

be present with such an approach. As we will demonstrate below, this approach does not increase the sensitivity so much that extremely high VCD values (several hundreds - thousands DU) can be retrieved, but instead increase the ability of the linearized approach to resolve moderate to high VCD values (perhaps tens - low hundreds DU).

## 2.5 Background State Construction

At every stage of the retrieval, the background state of the volcanic $SO_2$-free atmosphere must be accurate in order for this linearized method to succeed. Consequently, calculating accurate statistics of the background state is paramount. Each CrIS instrument collects almost 3 million spectra per day, allowing for robust characterization of the background spectrum including variation in conditions across seasons and locations. Because we use real CrIS spectra, not only are meteorological variations (including water vapor, clouds, temperature, etc) accounted for, but the instrument noise profile is also included.

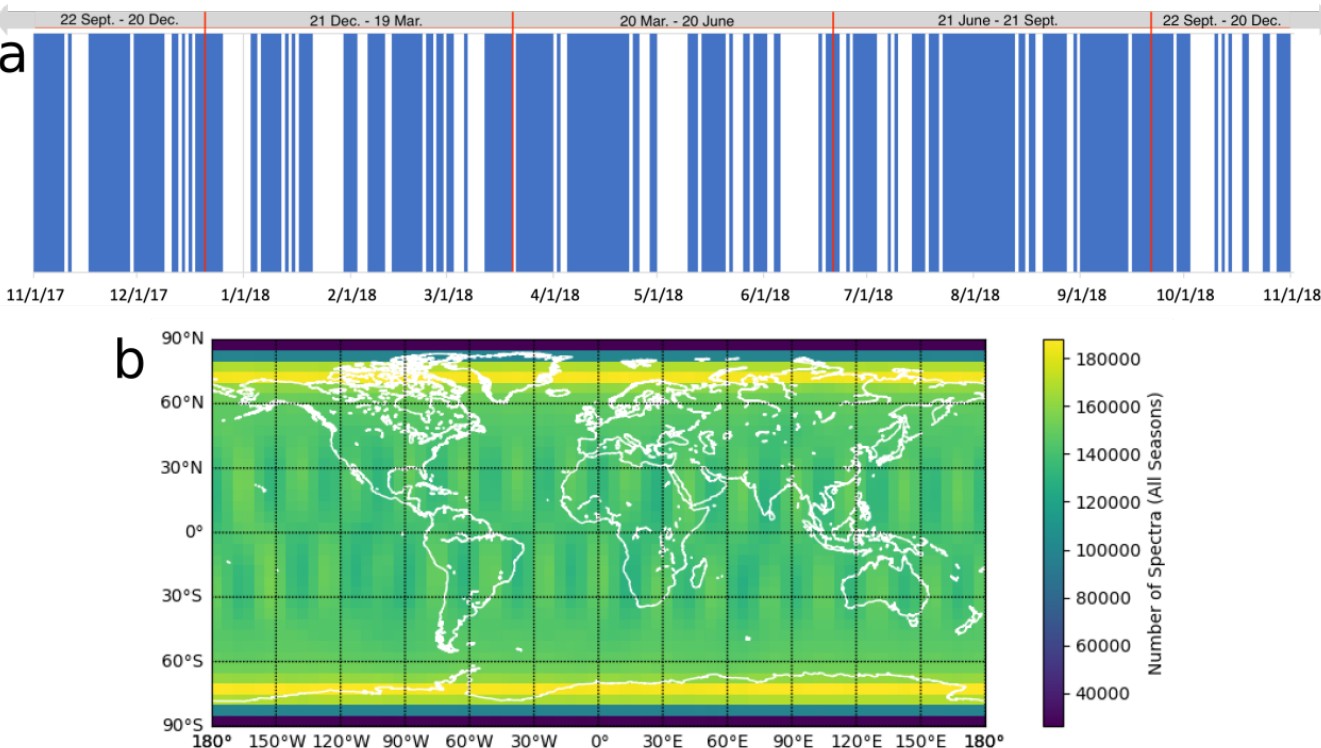

**Figure 3.** : *a) IASI-derived days with (blue) and without (white)* $SO_2$ *columns with VCD* $\geq 1$ *DU in the one year background construction interval (1 Nov. 2017 - 1 Nov., 2018). Date intervals across top define seasons. b) Histogram showing the number of SNPP CrIS spectra in each latitude, longitude cell totaled over the one year interval.*

In constructing the background spectrum channel-wise marginal PDFs (histograms) and covariance matrix, periods with little or no $SO_2$ must be determined. We utilize the detailed record of global volcanic $SO_2$ emissions from the operational IASI

SO$_2$ retrieval algorithm (L. Clarisse, *pers. comm.*) between 1 November, 2017 and 1 November, 2018, collecting all SNPP
CrIS spectra measured on days with maximum VCD less than 1 DU SO$_2$ present anywhere in the atmosphere (Fig. 3a).

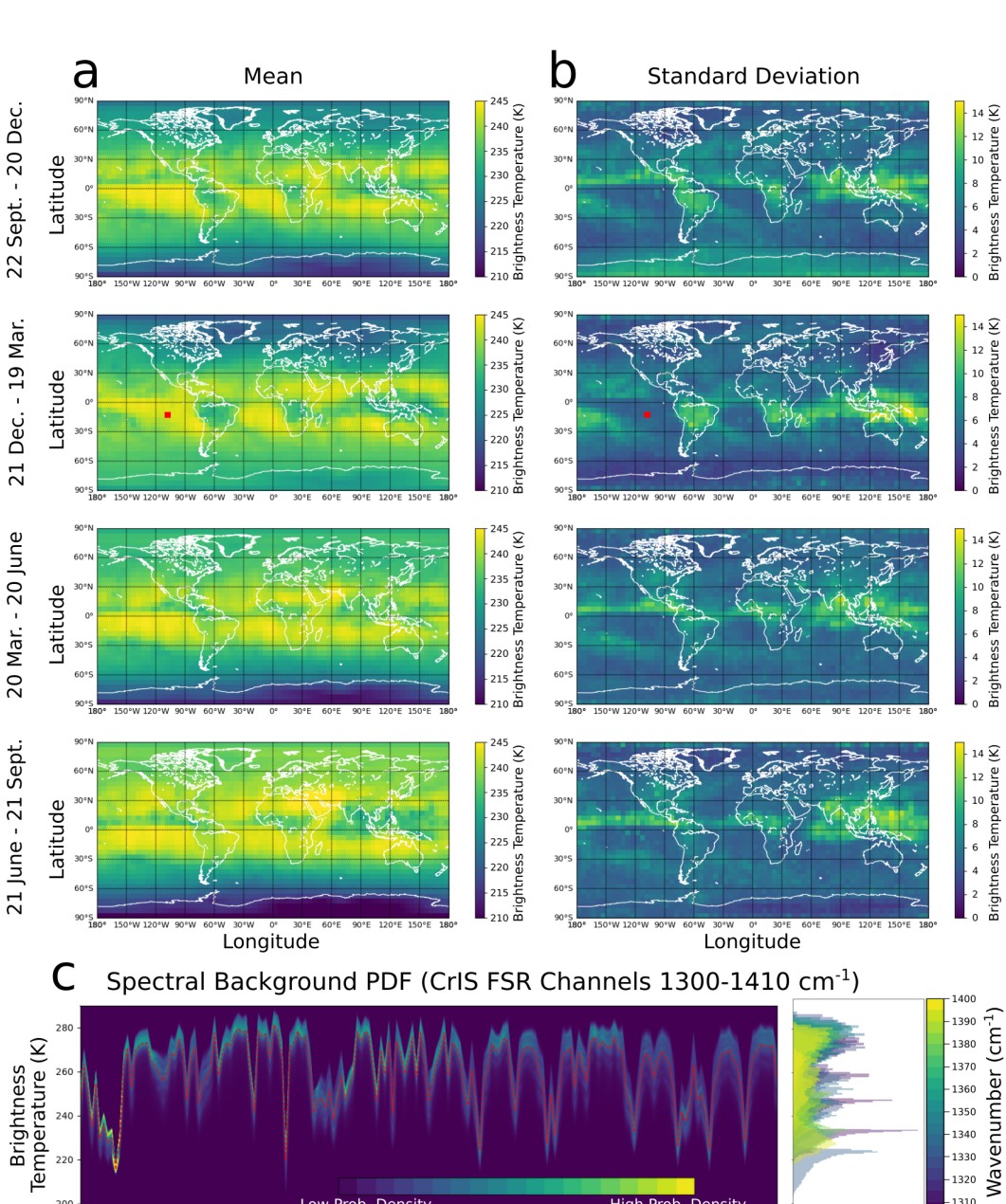

**Figure 4.** : *SNPP CrIS mean (a) and standard deviation (b) brightness temperature at 1300 cm$^{-1}$ for each grid cell and each season interval. The red grid cell corresponds to the data shown in (c). c) Marginal PDF of the background spectrum indicated by the red cell in (a,b) with mean spectrum (red dashes) and several individual marginal PDFs (right) shown.*

This leaves a database of more than $3.6 \times 10^8$ SO$_2$-free CrIS spectra over the one year period. We classify the spectra regionally and seasonally, partitioning this database into four seasons and $5° \times 5°$ latitude and longitude grid cells yielding 10,368 bins (Fig. 4). Each bin has a full set of marginal PDFs (brightness temperature histograms) for each channel and a covariance matrix characterizing the correlation structure among the channels. This partitioning reduces the the overall variability represented in the mean spectra and thus also reduces the magnitude of the error covariance matrix entries while still capturing the fundamental variability due to spectral trends between regions and throughout a year.

For each season-latitude-longitude bin, we construct a representative sample of 10,000 possible background spectra which conform to the set of channel-wise marginal distributions and covariance matrix relevant to that bin. Although our database is large enough to construct this sample for each bin, we generate these possible spectra by another method because it is preferable (from a mathematical standpoint) that the samples represent only what is known statistically about the spectra. That is a subtle point, but because the channel marginal distributions are represented as histograms (with finite range), generating synthetic background spectra very slightly damps the possible variance contained in a set of real measured spectra and limits the possibility of two key issues: (i) that real but anomalous or erroneous background spectra will be used and (ii) that real spectra with SO$_2$ just less than 1 DU (the database threshold) will be used as a supposedly SO$_2$-free background. For example, if extreme record-breaking conditions (e.g., hurricanes, droughts, etc.) appear in the database collection interval, their spectra will get caught in the database. These events will affect the covariance matrix and marginal distributions; however, they will cause far more variance in the retrieval if used as backgrounds than if they can only affect the used backgrounds as a forcing on the bin statistics. Additionally, construction of similar databases for other sensors could still proceed with fewer database entries since the statistics of the season-latitude-longitude bins would be expected to converge after fewer entries than were used here.

Since the channel-wise marginal distributions are generally non-Gaussian (e.g., Fig. 4), sampling the random vector $\boldsymbol{Y}_{bg}$ with covariance matrix $\mathbf{S}$ is non trivial. The general problem of sampling a correlated random vector with known non-normal marginal distributions and covariance matrix is accomplished by a transform sampling technique known as NORTA (NORmal To Anything) (Cario and Nelson, 1997). The NORTA process by which we generate samples of $\boldsymbol{Y}_{bg}$ is detailed in Appendix A. In the above retrieval (in the pre-screening and fully probabilistic phases), the background spectrum and covariance are interpolated spatially from the collection of binned backgrounds to each CrIS FOV center by a bilinear interpolation scheme using the four nearest season-latitude-longitude cells (Appendix C).

NOAA-20 CrIS has very similar radiometric characteristics as SNPP except that NOAA-20 FOV 7 noise is within specification and is therefore considered in this study (JPSS CrIS SDR Team, https://www.star.nesdis.noaa.gov/jpss/documents/AMM/N20/CrIS_SD Consequently, we use the SNPP-generated backgrounds in SO$_2$ retrievals with both SNPP and NOAA-20 CrIS spectra.

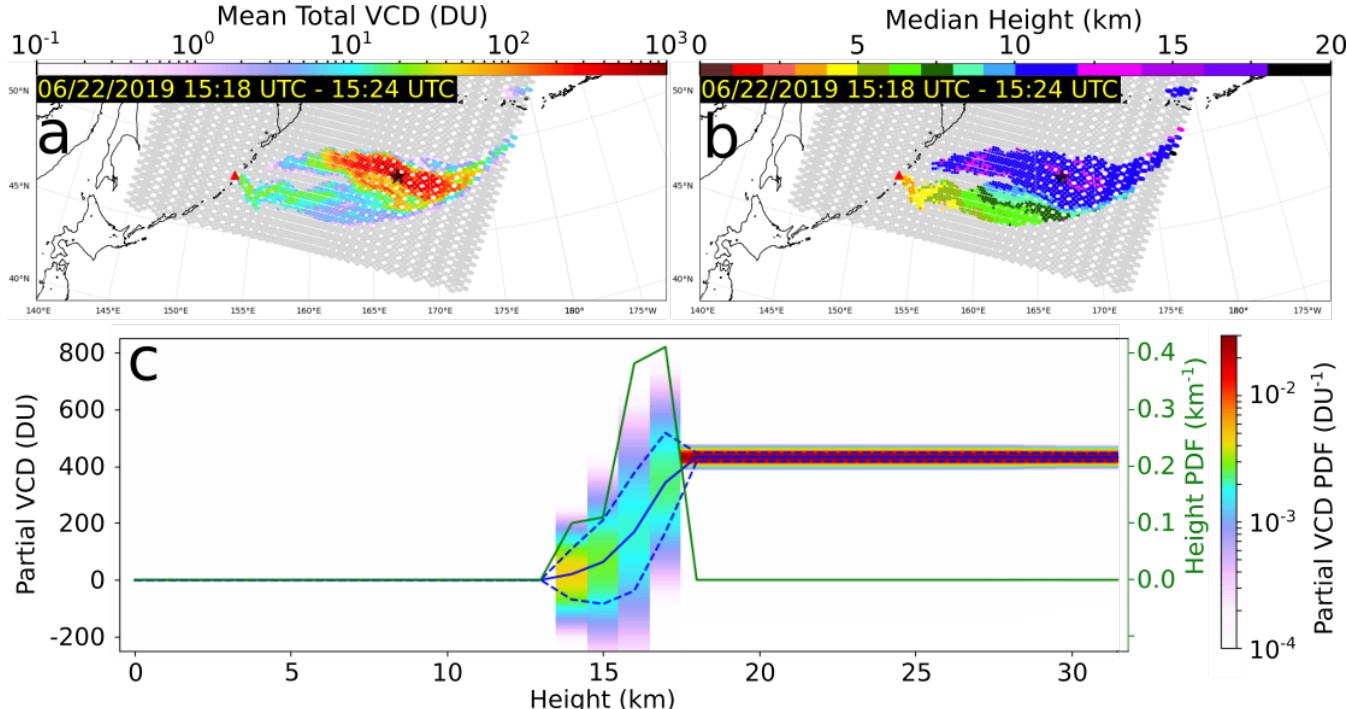

**Figure 5.** : *NOAA-20 CrIS mean total VCD (a) and median height (b) early in the evolution of the Raikoke eruption cloud. Star indicates location of (c). c) Probabilistic retrieval of $SO_2$ layer altitude (PDF, green) and partial VCD (mean, blue solid; mean $\pm$ standard deviation, blue dotted; PDF, color bar) for the strongest individual VCD measured by CrIS in the Raikoke cloud (48.52107 N, 167.25615 W, 22 June, 2019, 15:22:25 UTC).*

## 3 Results

### 3.1 Test Case I: Raikoke, Kuril Islands, 2019

At approximately, 18:00 UTC on June 21, 2019 (4:00 AM local time), Raikoke volcano in the Kuril Islands erupted for the first time since 1924 (Sennert, 2019 (Global Volcanism Program); Hedelt et al., 2019). The strongest pulses of the eruption rose to an altitude of approximately 13 km, forming an umbrella cloud which was quickly advected to the east by strong winds. In the first hours of the eruption, $SO_2$ columns with VCD > 900 DU (Hedelt et al., 2019, >1000 DU, S. Carn, *pers. comm.*) were detected. The strongest individual measurement made by our method (48.52107 N, 167.25615 W, 15:22:25 UTC, 22 June, 2019) had a mean total VCD of 432 DU with standard deviation of 15 DU (Fig. 5); however, because there is significantly greater uncertainty within the support of the height PDF, the largest value of mean plus uncertainty occurs just below the upper end of the height PDF support (Fig. 5c). This underestimate early in the Raikoke cloud history was likely due two factors, channel saturation despite the specialized strong column retrieval and the fact that the footprint and layout of CrIS FOV leaves many gaps ($\approx$30% by area) where extremal values could have been present. Because this analysis focuses on the $SO_2$ $\nu_3$ band

(1300-1410 cm$^{-1}$), it is very unlikely that ash was responsible for the strong underestimation (e.g., Carboni et al., 2012).

As described above, more specialized retrievals could be devised to increase the sensitivity to very strong $SO_2$ loading as in

the Raikoke case; however, such schemes are beyond the scope of this work, which is principally concerned with advances in height information and estimating VCD values in more typical (low-moderate concentration) emissions. Within about one day, the ash and $SO_2$ were entrained into a large extratropical cyclone which heavily distorted the dispersion of the cloud, with the $SO_2$ cloud being pushed to the north and dispersing in both easterly and westerly directions (Fig. 6). Early in this complex dispersion, $SO_2$ VCD values remained strong despite a rapid decline in eruptive output. This is most likely a result

of the convergence caused by entrainment into the cyclone. Based on the probabilistic retrieval and tropopause data from the National Centers for Environmental Prediction (NCEP), it is clear that the vast majority of the $SO_2$ cloud mass was in the lower stratosphere, with only a small lower layer in the mid-upper troposphere which had mostly dispersed after the first week (Fig. 6b-l). After one month, the $SO_2$ cloud had spread out over most of the northern hemisphere above $30°$ N with most VCD values $< 2$ DU; however, some columns remained as strong as 20 DU. After two months, traces of the $SO_2$ cloud remained

over Northern Canada and the Hudson Bay with all measured VCD less than 1 DU.

### 3.2 Test Case II: Early Detection of $SO_2$ Emission from Bogoslof Volcano, Aleutian Islands, 2016

In the 2016 - 2017 eruptive period at Bogoslof volcano, 70 explosive events were identified (Coombs et al., 2018, 2019). The first five explosions were not detected in real time, and could only be identified and characterized after reanalysis of satellite and other data sources (Coombs et al., 2019). The first CrIS detection of the $SO_2$ cloud from this sequence of explosions

occurred at UTC 22:48 on 16 December, 2016 (cluster of 17 CrIS FOVs, approximately 300 km NE of Bogoslof), which was most likely the Event 4 (UTC 18:39) $SO_2$ plume drifting downwind (Coombs et al., 2018, 2019). As noted in Coombs et al. (2018), the USGS Alaska Volcano Observatory (AVO) was not able to issue a Volcanic Activity Notice (VAN) for this small event and consequently no height information was generated until the reanalyses of Schneider et al. (2020) in which a cloud height of 6.1 km was determined. The SACS near-real time retrieval (https://sacs.aeronomie.be/) only detected $SO_2$ from this

cloud in two IASI FOVs which was not sufficient to trigger an alert notification. This small pulse was not observable by the multispectral infrared remote sensing methods nor by automated analysis of multi-spectral signatures and cloud growth rates (Pavolonis et al., 2013, 2015a, b, 2018; Schneider et al., 2020). CrIS median heights are mainly clustered between 5 - 8 km with some scatter due to localized cloud edge effects (Fig. 7c,d). This is broadly consistent with the reanalysis of Schneider et al. (2020).

Of particular importance in this small cloud made up of only a few (17) FOVs, SNPP CrIS FOV 7 is significantly nosier (above specification) than that of other FOVs (Zavyalov et al., 2013; Han et al., 2013) and consequently, the FOV 7 retrievals are highly suspect and have not been used to estimate the $SO_2$ height. The are included in Fig. 7d,e mainly to illustrate how an increase in instrument noise affects height information. As might be expected, increased instrument noise propagates larger uncertainty to the height PDFs and tends to distribute their centers to the lower and upper end of the range of considered

altitudes. This is the signature of the central role the $\arg\max$ operation plays in the retrieval. For example, if higher noise is

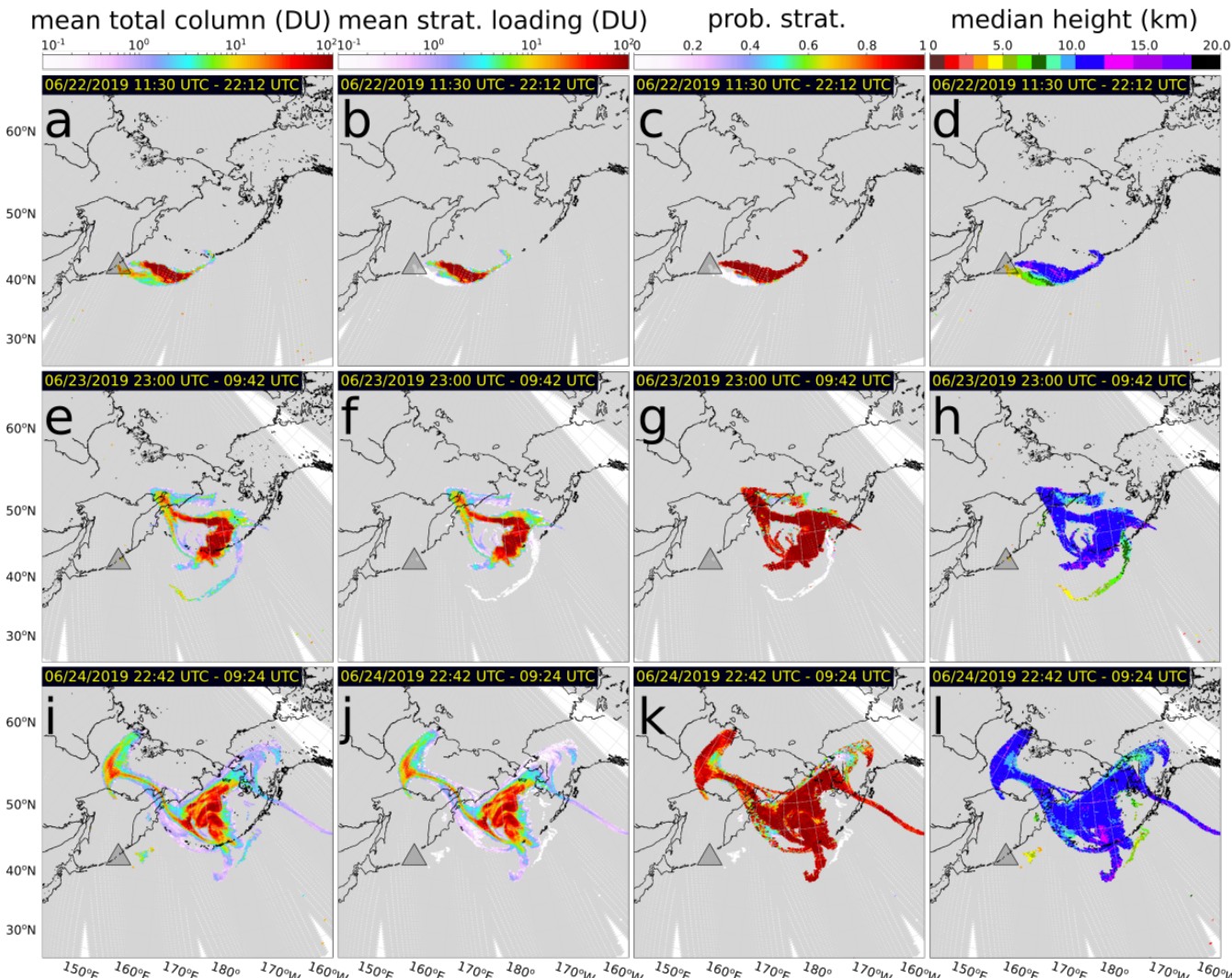

**Figure 6.** : *Time evolution (top to bottom) of the Raikoke SO$_2$ plume from NOAA-20 CrIS showing the expected value total VCD (a,e,i), the expected value stratospheric partial VCD (b,f,j), the probability that the SO$_2$ layer is in the stratosphere (c,g,k) and the median layer height (d,h,l). The height of the tropopause was calculated from daily NCEP reanalysis data.*

propagated to the z-score height functions, the $\arg\max$ can produce wildly different heights even for small differences in the z-score height profile.

Because the probabilistic framework allows the calculation of a mean partial VCD, we may derive a formula for the mean or expected concentration profile by similar means as for Eq. 15:

$$410 \quad \mathbb{E}[C(h)] = \mathbb{E}[\frac{d}{dh}\hat{X}(h)] = f_H(h)\,\mathbb{E}[\hat{X} \mid H{=}h] \tag{18}$$

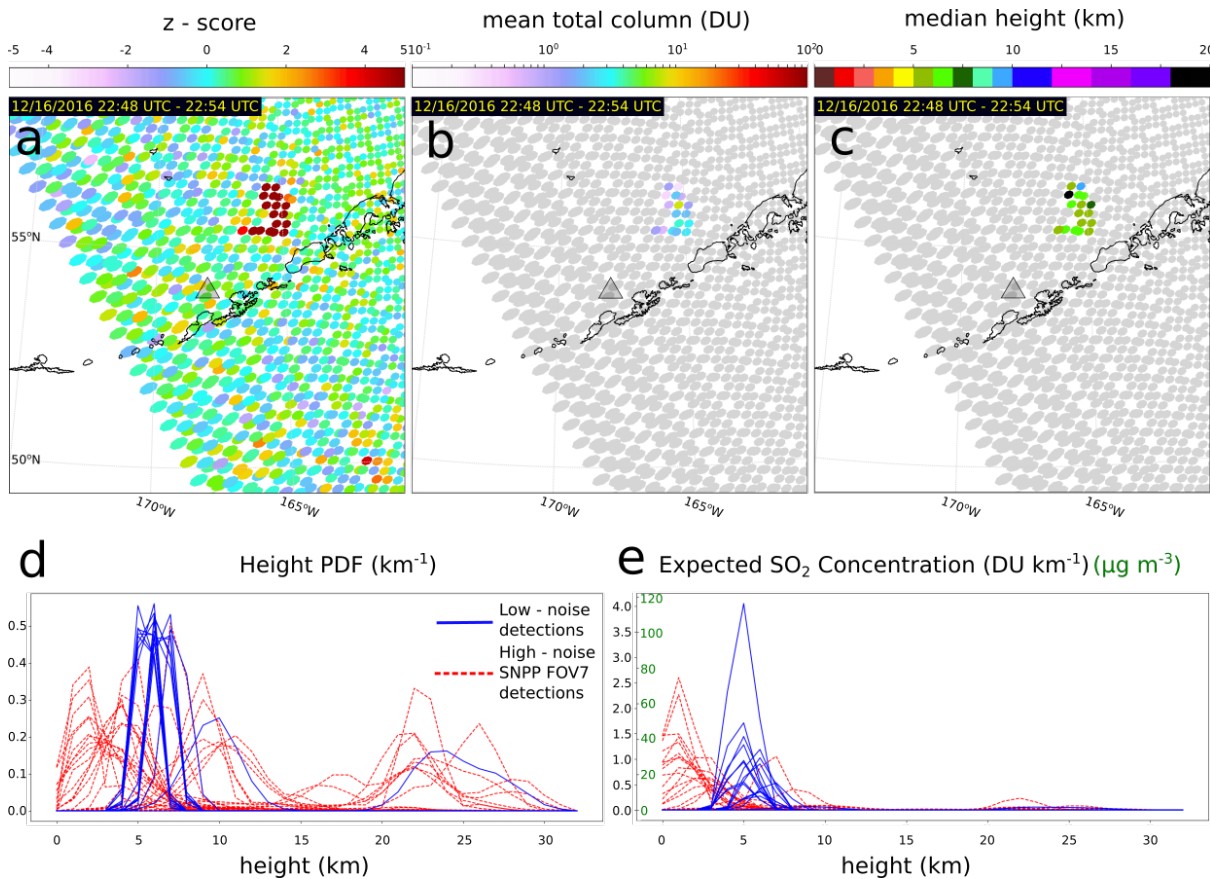

**Figure 7.** : *Upper: Initial (classical) z-score (a), mean total VCD (b), and median height (c) for an explosion from Bogoslof very early in the 2016 - 2017 eruption (SNPP CrIS). Lower: Height PDFs (d) and expected (mean) cloud concentration (e), mean concentration profiles for the detected $SO_2$ cloud. Retrievals from the high noise SNPP CrIS FOV 7 (red dotted in lower panel) are not shown in (a,b,c).*

which is shown for FOVs in the detected Bogoslof cloud (Fig. 7e). This example demonstrates that the CrIS $SO_2$ detection and characterization scheme is sufficiently sensitive to capture some small emissions which are generally difficult to observe by other means.

### 3.3 Test Case III: Resolving Strong Stratification in an $SO_2$ Plume, Sierra Negra, Galapagos Islands, 2018

On 26 June, 2018 after a period of elevated seismicity, the onset of a major eruption at Sierra Negra was signaled by volcanic tremor at 19:40 UTC, producing an ash and $SO_2$ plume at 20:09 UTC (Carn et al., 2018; Vasconez et al., 2018; Hedelt et al., 2019). The first CrIS observation also occurred at 20:09 UTC from SNPP, detecting $SO_2$ above the Sierra Negra on 3 adjacent FOVs on the edge of scan with maximum initial z-scores of approximately 9, 16, and 31. Subsequent overpasses show the plume rising to approximately 14 - 19 km and spreading in a complex manner due to vertical wind shear as evidenced by the

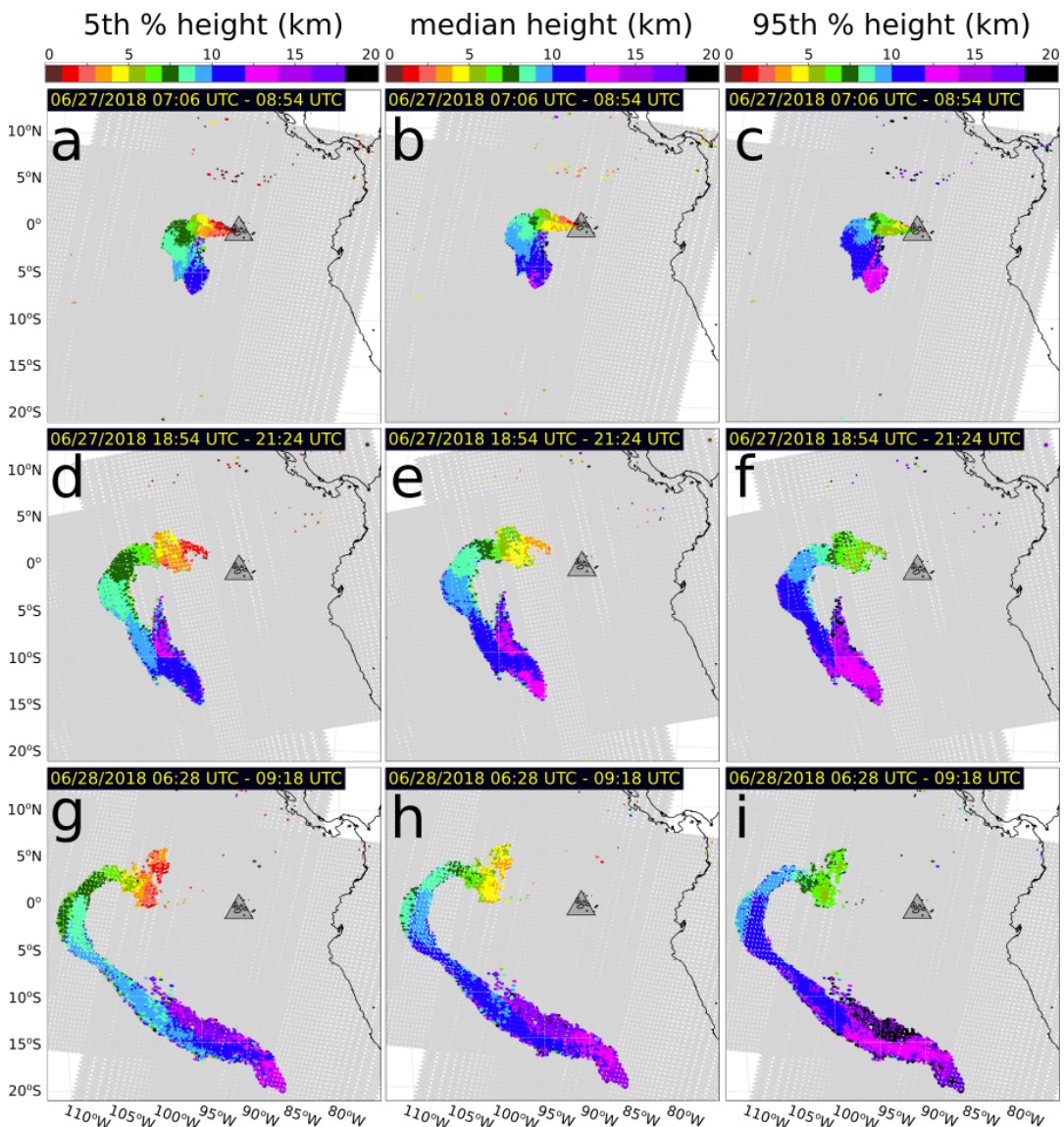

**Figure 8.** : *Time evolution (top to bottom) of the 27 June, 2018 Sierra Negra $SO_2$ plume height represented as the $5^{th}$-percentile (a,d,g), median (b,e,h), and $95^{th}$-percentile (c,f,i) heights. Data is merged SNPP CrIS and NOAA-20 CrIS with SNPP CrIS FOV 7 excluded.*

lower plume altitudes spreading towards the west and the upper plume altitudes spreading towards the southeast (Fig. 8). The significant shearing of the eruption plume enables direct observations of the cloud at many levels. Consequently, this eruption forms a good opportunity to highlight the broad sensitivity of this method in detecting and characterizing $SO_2$ at every elevation from the vent (1.124 masl) up to $\sim$14 - 19 km and potentially higher. Additionally, this example highlights the strength of the probabilistic height retrieval, enabling the retrieval of any desired confidence interval on the height. Here we compute the

90% confidence interval (Fig. 8), highlighting the fact that the 95th- and 5th- percentile are not in general symmetric about the median nor are the same size at different locations within the plume. Because our method retrieves consistent statistics across all measurements, we can ensure the stability of the method and derived probabilities in particular, giving good smoothness even without post-processing. As described above this is not necessarily the case for other height retrievals which compute a single estimate with constant uncertainty.

# 4  Discussion

## 4.1  Comparison with other data

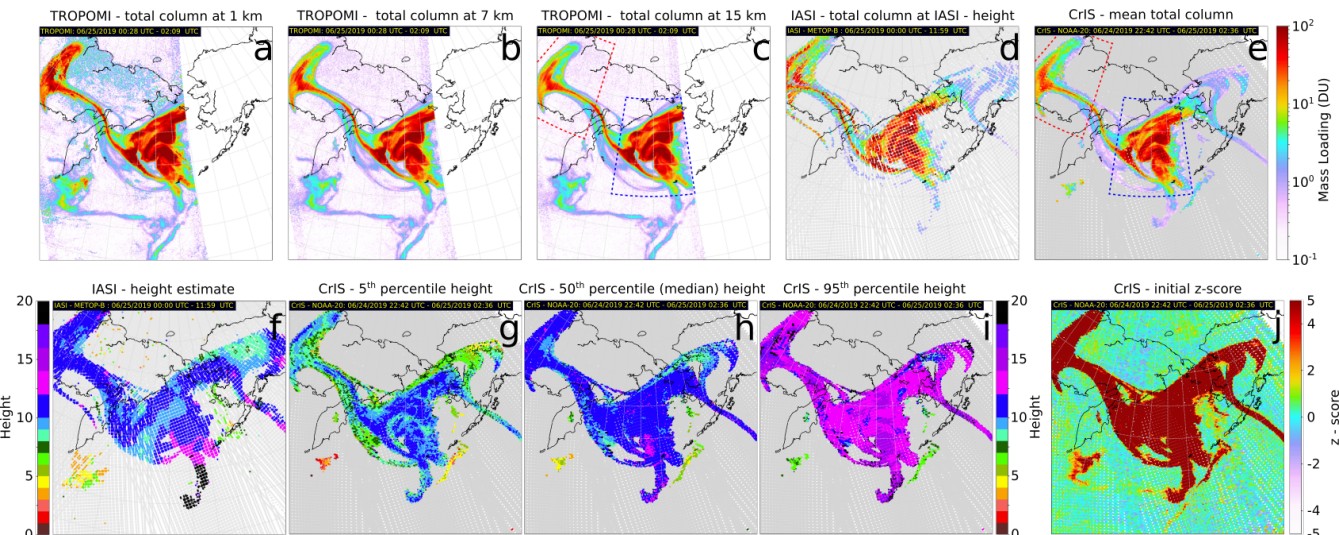

**Figure 9.** : *Representative comparison between TROPOMI, IASI, and CrIS $SO_2$ data. Top: TROPOMI VCD given a 1 km-thick layer at 1 km (a), 7 km (b), and 15 km (c) altitudes; IASI (METOP-B) VCD given a 1 km-thick layer at the IASI height estimate (f); and CrIS (NOAA-20) mean total VCD (integrated against the height PDF). Bottom: IASI height estimate (f); CrIS $5^{th}(g)$-, $50^{th}$(median, h)-, and $95^{th}(i)$- percentile heights; CrIS initial z-score (j). The CrIS-TROPOMI comparison regions analyzed in Fig. 10 are shown in (c),(e) as red and blue dashed outlined latitude-longitude boxes.*

Although a deep analysis of the differences between the present method and others is beyond the scope of the present work, here we highlight a brief, representative comparison of our $SO_2$ retrievals with data from TROPOMI, IASI, and the Cloud-Aerosol Lidar with Orthogonal Polarization (CALIOP) during the evolution of the Raikoke eruption cloud. These sources of 435  data are described briefly here.

As mentioned above, CrIS is a very similar instrument to IASI (both Fourier transform Michelson interferometers). The relevant instrument differences are that IASI (aboard EUMETSAT satellites METOP-A/B, LTAN: 9:30 PM) covers both the $\nu_1$ and $\nu_3$ $SO_2$ absorption bands (though only $\nu_3$ is used to generate IASI heights), IASI's spectral resolution is 0.5 cm$^{-1}$

(apodized spectra) compared with 0.625 cm$^{-1}$ for apodized FSR CrIS, and as mentioned above, despite IASI's slightly smaller

FOV size (12 km-diameter at nadir) compared with CrIS (14 km-diameter at nadir), the greater spatial number density of CrIS
FOVs gives CrIS a higher resolution than IASI by virtue of a smaller average sampling distance. For comparison with the set
of NOAA-20 CrIS overpasses of the Raikoke cloud in Fig. 9 (6/24/2019 22:42 to 6/25/2019 02:36 UTC), the nearest IASI
height data were collected from IASI instrument aboard METOP-B between 6/25/2019 00:00 to 11:59 UTC. Although some
of the data in this interval are highly asynchronous, the cloud did not experience major changes in this time and the heights

(and to a lesser extent VCD) can be compared. As described in Table 1, IASI heights are generated by the Clarisse et al. (2014)
retrieval and the VCDs are computed by a related nonlinear technique using the retrieved heights (Clarisse et al., 2012). In the
framework presented here, this VCD is the conditional VCD sampled at the IASI-retrieved height.

TROPOMI is a UV spectrometer operating aboard the TROPOMI the Copernicus Sentinel-5 Precursor (S5P) satellite which
orbits only 3.5 minutes behind SNPP CrIS (Veefkind et al., 2012). TROPOMI represents a significant advance in monitoring

of SO$_2$ and other atmospheric constituents due to its very high spatial resolution ($7 \times 3.5$ km$^2$ pixels at nadir) increasing the
sensitivity and signal to noise ratio in a given region by a factor of 3 over OMI ans OPMS (Theys et al., 2017, 2019; Fioletov
et al., 2020). The TROPOMI SO$_2$ data used here are generated from back-scattered UV radiances with the S5P operational
processing algorithm (a differential optical absorption spectroscopy (DOAS) technique)(Theys et al., 2017). Because most
UV-based techniques are not able to retrieve height information, UV-based VCD data are typically presented as conditional

VCD for several heights. The TROPOMI SO$_2$ conditional VCDs are given for assumed SO$_2$ plume altitudes of 1 km, 7 km,
and 15 km. As the vast majority of the Raikoke cloud was in the stratosphere, we use the 15 km data for direct comparison
with CrIS (Fig. 10). Obviously such high-resolution data is difficult to compare directly with CrIS, so it was first resampled to
the CrIS FOV footprints by constructing weighted averages of TROPOMI pixel data with weights determined by the fraction
of intersecting area between the pixels and each elliptical CrIS FOV, as in Sun et al. (2018). Unfortunately, since SNPP CrIS

was experiencing a major outage during the Raikoke cloud's evolution due to an electrical fault, only NOAA-20 CrIS was able
to measure the cloud and consequently, the NOAA-20 CrIS-TROPOMI comparison is asynchronous by about 50-55 minutes
instead of the 3.5-5 minutes that would have been possible if SNPP CrIS was functioning.

The last source of comparison data is 532 nm backscattered lidar measurements from the CALIOP overpass of the Raikoke
cloud between 14:32 and 14:36 UTC on 25 June, 2019 (Fig. 11). CALIOP profiles aerosols, clouds, and other features between

the ground and lower stratosphere (Winker et al., 2009). Although it cannot detect molecular SO$_2$, it can detect volcanic ash as
well as sulfate aerosols which are a photo-chemical product of SO$_2$ in the atmosphere. As is typical of volcanic SO$_2$ studies,
highly attenuating CALIOP aerosol layers (especially in the stratosphere) are considered here as a proxy for the presence of
SO$_2$ (Clarisse et al., 2014; Carboni et al., 2016, e.g., ).

As mentioned above, our strongest total VCD measurement from the Raikoke cloud was 432 DU. This is significantly lower

than the maximum detected by TROPOMI ($> 900$ DU, (Hedelt et al., 2019)), and several other UV-based methods ($\sim$1000
DU, S. Carn, *pers. comm.*). This suggests that our method, despite the integrated height estimate and the specialized retrieval
for strong SO$_2$ loading, currently cannot fully capture these extremely high VCD values; however, away from these extreme
values, our retrieval performs well in comparison to TROPOMI and IASI (Fig. 9 a-e, Fig. 10a-e). Other than the relatively few

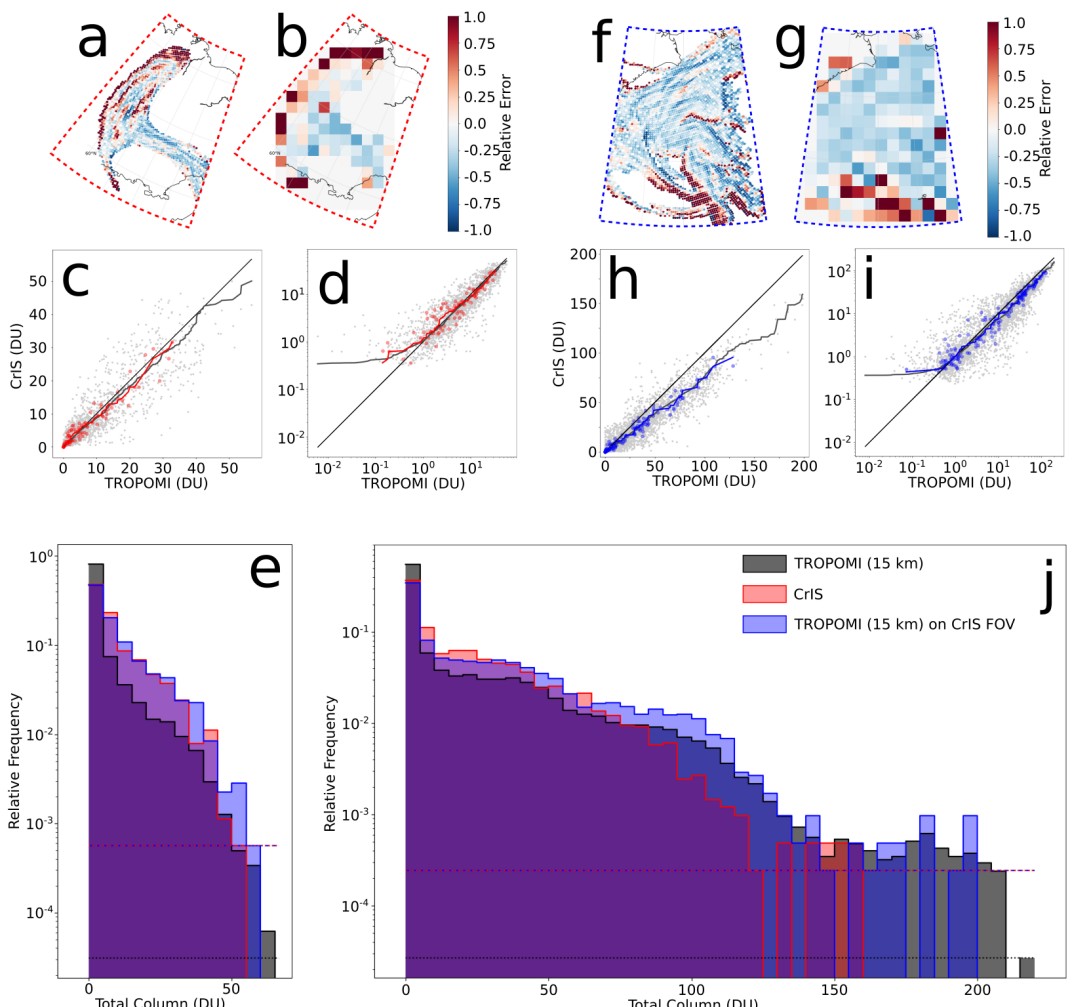

**Figure 10.** : *Comparison between TROPOMI and CrIS SO$_2$ data for the red (a-e) and blue (f-j) regions outlined in Fig. 9. a) Relative difference between CrIS FOVs and TROPOMI-synthesized CrIS FOVs for red-outlined region. b) Same as (a), but smoothed onto a 100 km × 100 km grid. c) direct (dots) and rank-order (lines) comparison of CrIS and TROPOMI VCD from (a;grey) and (b; red). d) log-scale version of (c). e) histogram (relative frequency) comparison for CrIS and TROPOMI data in red-outlined region with dotted lines corresponding to the relative frequency of a single measurement. f-g) same as (a-e) for blue-outlined region.*

columns with unusually large VCD values, the largest discrepancy between the TROPOMI-retrieved Raikoke cloud and that from CrIS is that the CrIS retrieval does not fully resolve the long, narrow, and diffuse east-west-running cloud to the south of the main cloud, although the CrIS retrieval does resolve some similar features elsewhere in the cloud. This may be due to the very high spatial resolution and sensitivity of the TROPOMI data compared with the coarse CrIS resolution which also contains ≈30% gaps. An additional contributing factor is that the CrIS retrieval starts with an initial detection of the z-score for pre-screening (Fig. 9 j) and only retrieves the height and VCD for FOVs with $z > 5$. Traces of this narrow, diffuse cloud

are present in the initial z-score field, although it mostly presents with a z-score below the detection threshold but slightly above the background noise. Because the western part of this cloud is low altitude, the weak signal in CrIS and IASI but good detection by TROPOMI is evidence that the IR sensors were limited somewhat by interference with larger quantities of water vapor. Overall it is likely that all of these factors played a role.

A more detailed view of the comparison between CrIS VCDs and TROPOMI conditional VCDs at 15 km altitude is shown
for two regions of the Raikoke cloud in Figure 10. As described above, these measurements are asynchronous by about 50-55 minutes, which is evident in Figure 10a,b,f,g where the largest errors between CrIS and the CrIS-resolution TROPOMI occur at the cloud edges (as well as internally in areas with more complex motion) where it had clearly moved between these observations. These two regions were selected because they both contain VCDs spanning several orders of magnitude but different ranges. The northwestern region (red-dashed, Fig. 9, 10a-e) contains VCD values generally up to about 50 DU
whereas the southeastern region (blue-dashed, Fig. 9, 10f-j) contains VCD values as high as 230 DU. As can be seen from the histograms of native resolution and CrIS-resolution TROPOMI, the lower spatial resolution (and gaps) of CrIS is at least partially responsible for its inability to resolve some of the highest VCD values measured by TROPOMI in native resolution. In regions of the cloud with generally lower concentrations CrIS and TROPOMI compare well, scaling approximately one-to-one with much of the noise being generated by the asynchrony. Although not a perfect comparison, the rank-order correlation is
also shown for the CrIS-resolution comparison and a coarse, 100-km pixel aggregate. Taken together, the FOV-by-FOV and 100 km pixel comparisons (both exact matching and rank-order) demonstrate that CrIS matches TROPOMI VCDs moderately well for these low-moderate concentration clouds. In higher concentration clouds (Fig. 10f-j), the comparison has a similar noise profile; however, CrIS consistently underestimates TROPOMI by about 75% ($-0.25$ relative error, Fig. 10f,g). Based on the histograms and correlation of CrIS and CrIS-resolution TROPOMI, there is a similar distribution of VCDs below about 50
DU, becoming very different by about 100 DU. In all regions of the Raikoke cloud, the sensitivity of the fully probabilistic retrieval is limited to VCDs greater than about 0.3 - 0.5 DU.

The main focus and strength of our approach is the ability to generate physics-based PDFs for the height. Because our retrieval is based on the operational algorithm in use for IASI, our retrieved heights are very similar to those from IASI although there are key differences readily apparent in Figure 9f-i. As mentioned above, the IASI heights represent the height
retrieved due to the mean background spectrum; however, because the retrieval of height is not linear (due to the $\arg\max$ operation), the retrieved height is not the expected value height. Inspection of the PDFs generated by this approach show that they are typically non-symmetric, although exact comparison is not possible due the orbital separation between the satellites carrying IASI (METOP-A,B) and those carrying CrIS (SNPP, NOAA-20). At least for the Raikoke cloud, the IASI heights are almost entirely bound within the CrIS 90% confidence interval (Fig. 9 f,g,i) and are similar to the CrIS median heights
(Fig. 9 h) in most areas. For the snapshot of the Raikoke cloud shown in Fig. 9, the largest differences in height appear at the southernmost part of the cloud, where the IASI heights are $> 19$ km altitude over a significant region. Furthermore, the IASI height estimate (Fig. 9 f) varies significantly over nearby, continuous parts of the cloud, whereas the CrIS median height is more consistent across space with some minor variation due to noise.

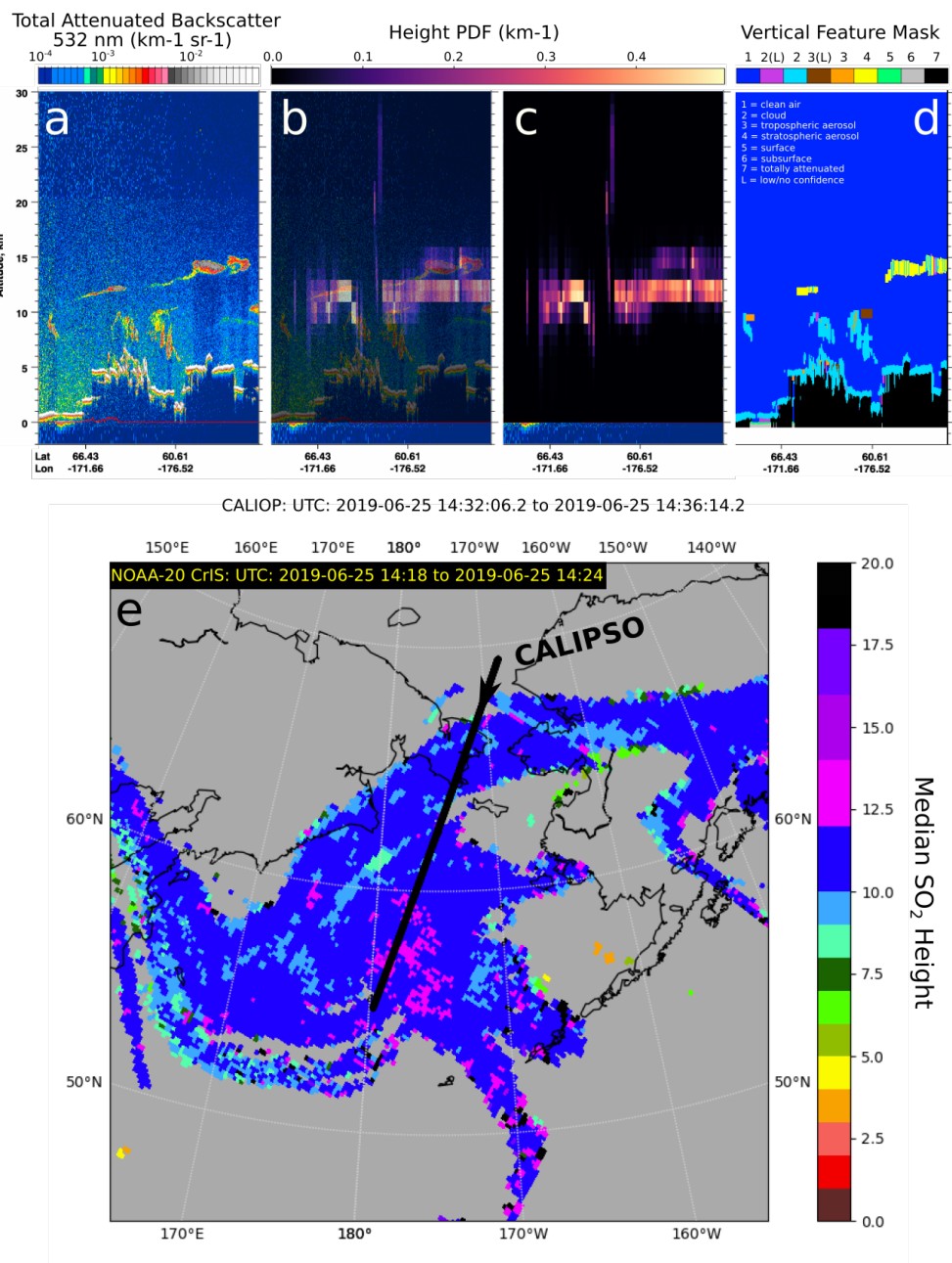

**Figure 11.** : *Representative example comparison of CALIOP lidar backscatter (a, b), CrIS* $SO_2$ *height PDFs (b-transparent, c), and CALIOP vertical feature mask (d) for the Raikoke cloud on 25 June, 2019. e) Nearest neighbor gridded interpolation of CrIS* $SO_2$ *median heights with closest CALIOP overpass shown (black, arrow indicating descending orbit,* $< 15$ *minutes after CrIS acquisition).*

Although not shown here, Hedelt et al. (2019) have recently developed a new $SO_2$ height retrieval for TROPOMI using inverse learning machines. Although computationally expensive to train, such an approach has the advantage of computation speed of the inversion once deployed, though it has not yet been incorporated into the TROPOMI SO2 data product as of this writing, so a direct comparison is not possible. However, it is clear from the data presented in Hedelt et al. (2019) that their height estimates for the Raikoke cloud span the CrIS 90% confidence interval, though they are significantly nosier than CrIS $SO_2$ height estimates and display a very prominent negative trend in height versus VCD, leading to systematically higher layer heights on the cloud edges than in the cloud centers (Fig. 14 of Hedelt et al., 2019). CrIS PDFs very rarely show similar trend.

Because we retrieve a PDF on each CrIS FOV rather than a single estimate, we can compare the PDFs directly to data from a CALIOP overpass of the cloud. Here we show an example comparison from Raikoke; however, a full comparison for every overpass of the Raikoke cloud is the subject of future work. For the first several days after the eruption, there was still significant ash suspended in the dispersing cloud, leading to the appearance of several highly attenuating layers in CALIOP data between 10 - 15 km (Fig. 11 a,b). The comparison we focus on is between CrIS retrievals from 14:18:00 - 14:24:00 UTC and CALIOP data from a subsequent overpass between 14:32:06 - 14:36:14 UTC on 25 June, 2019. To prevent the mixing of nearby height PDFs, the CrIS data are interpolated to the CALIOP track by nearest neighbor interpolation, creating a profile of the nearest CrIS $SO_2$ height PDFs. The nearest neighbor interpolation over all of space is shown in Fig. 11e with 16 km pixels (the average CrIS sampling distance).

Overall, there is good agreement between the CrIS $SO_2$ height PDFs and the altitudes of the strongly attenuating CALIOP layers; however, the CrIS PDFs have several important characteristics that complicate comparison. There is some minor noise derived mainly from two anomalous CrIS FOVs at the cloud edge ($\approx 62°$ N, $175°$ W), exactly at the CALIOP track, leading to anomalously high altitudes there (Fig. 11 e,). Most interestingly, some regions of the cloud have bimodal PDFs (Fig. 11 b,c, south of $60°$N). Preliminary investigations of this retrieval suggest that such occurrences are not exceedingly rare throughout this and other clouds; however, such PDFs occur widely and tend to persist over moderate distances (as in Fig. 11 b,c).

In the most strict sense, such PDFs can only be attributed to the presence of similar statistical features of the background. Specifically, if the background spectrum probability space is dominated by two sets of meteorological conditions (for example, one mode representing deep convective cloud radiances and another for cloud-free radiances), then multiple populations of the Monte-Carlo height samples may accumulate, leading to a multimodal height PDF.

Relaxing this strict interpretation, there is some evidence that these bimodal PDFs may represent the presence of $SO_2$ at multiple altitudes. In this case, the strongest CrIS probabilities occur in a lower layer around 12 km altitude (Fig. 11 b,c) suggesting the presence of molecular $SO_2$ layer; however, there is no attenuating CALIOP layer there. Additionally, the CrIS retrieval assigns significant (but less) probability mass to a higher level collocated with a strongly attenuating CALIOP layer at about 15 km. Since this data is very early in the cloud's evolution ($<$5 days), this layer is likely dominated by volcanic ash rather than being dominated by sulfate aerosols, having not had enough time to convert large portions of the erupted $SO_2$. Of note, because the upper CALIOP layer is very strongly attenuating, it may completely shadow any evidence of lower diffuse ash clouds if they existed. Considering that ash minimally affects the $SO_2$ $\nu_3$ band, this collocation at 15 km altitude suggests that this strongly attenuating layer (likely mainly ash) contains $SO_2$. The fact that the stronger of the two probabilities is in

the lower layer combined with the CALIOP-CrIS collocation could be interpreted as evidence that there is $SO_2$ at both layers. Although this is a possibility, confirmation of such a configuration would require a deeper analysis including forward and inverse trajectory modelling with advanced data assimilation techniques and therefore is well beyond the scope of the present work.

## 4.2 Long-term analysis of the Raikoke $SO_2$ cloud

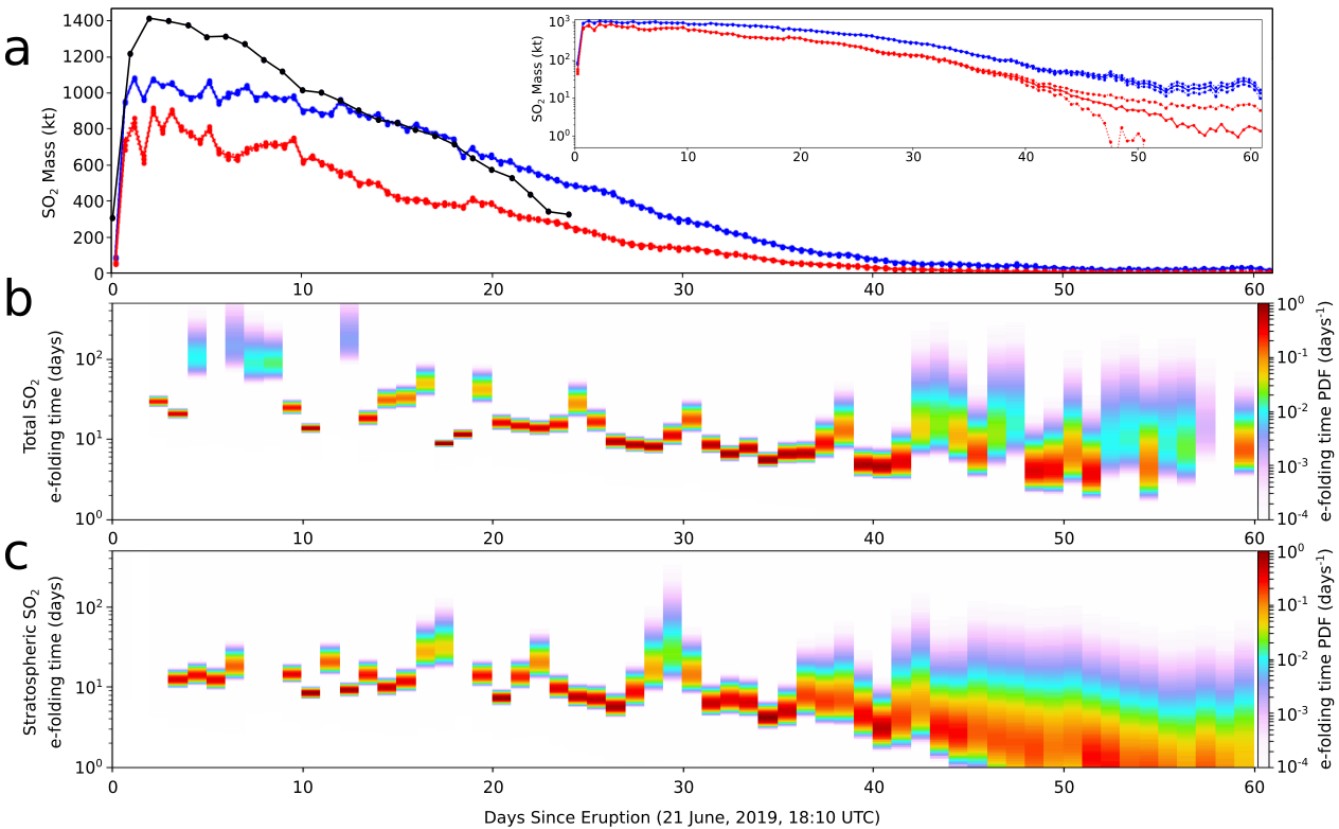

**Figure 12.** : *a) Total (blue) and stratospheric (red) mass of* $SO_2$ *above* $30°$ *N for two months following the eruption of Raikoke showing CrIS-derived mean (solid lines)* $\pm$ *1 standard deviation (dotted lines), log-scale inset. Black: preliminary TROPOMI mass (Fig. 12; NASA/GSFC, 2019, last access: 2020-08-18, https://so2.gsfc.nasa.gov/pix/special/2019/raikoke/raikoke_tropomi_so2stl_0621-07062019.html). "Instantaneous" (daily-aggregated) e-folding time PDFs for the total (c) and stratospheric (d)* $SO_2$ *in the Raikoke eruption cloud.*

By retrieving PDFs for height and partial VCD it is possible to enhance time series analysis of $SO_2$ clouds accordingly, enabling the generation of time series with quantified uncertainty. As an example, we calculate the total and stratospheric $SO_2$ mass time series probabilistically as sums of many independent retrievals (Fig. 12).

To estimate the mass of an $SO_2$ cloud, the values of the retrieval on the CrIS FOVs must be interpolated onto a continuous grid spanning the cloud. In the present study we use an equal-area grid and perform nearest neighbor interpolation of the CrIS FOV retrievals. We calculate the mass (with uncertainty) as in Appendix E for both the total $SO_2$ mass in the atmosphere and in the stratosphere only. The stratospheric partial VCD values $\hat{X}(h_{tropopause})$ were calculated from Eqns. 17a and 17b using daily NCEP/NCAR tropopause pressure level and geopotential height reanalysis data (Kalnay et al., 1996). By exploiting the fact that the mass is a normal random variable at each time, we can from these results estimate the daily "instantaneous" apparent e-folding time of the $SO_2$ as $\tau = -M(t)/\dot{M}(t)$ with $\dot{M}(t)$ calculated by finite difference and the PDFs computed by standard results in probability theory (Fieller, 1932; Hinkley, 1969; DeGroot and Schervish, 2012, Appendix E). The e-folding times shown here are really apparent measurments since mass is lost from the cloud not only due to photo-chemical conversion of $SO_2$ to sulfate aerosols, but is also lost due to the dispersion and dilution of the cloud below levels that can be detected. We include this time series because it illustrates some interesting aspects of the Raikoke cloud's evolution and because it illustrates a logical extension of how uncertainty is propagated through this work.

From Fig. 12 it is clear that the $SO_2$ cloud, as characterized by CrIS, did not immediately show the exponential mass decay that has been used in similar studies of large eruptions (e.g., Read et al., 1993; Carn et al., 2017; Krotkov et al., 2010). The observed trend is likely a combination of retrieval limitations and genuine atmospheric chemical properties. As described above, the CrIS VCD underestimate early in the Raikoke cloud history was likely due to channel saturation despite the specialized strong column retrieval. The strongest CrIS VCD were only approximately $50\%$ of the strongest columns reported at the time; 432 DU from CrIS vs. >900 DU (Hedelt et al., 2019) and >1000 DU (S. Carn *pers. comm.*) and in aggregate, CrIS total VCD were approximately $75\%$ of those of TROPOMI in the most concentrated regions (Fig. 10). This underestimate was propagated to the CrIS maximum total mass estimate of approximately 1.1 Tg $SO_2$ (Fig. 12) compared with preliminary estimates from other sources (1.4-1.5 Tg, Sennert, 2019 (Global Volcanism Program) , S. Carn *pers. comm.*). Very large VCD values persisted for many days though after approximately 10 days, most of the cloud had diluted sufficiently that CrIS and preliminary TROPOMI data were in good agreement between days 10 and 20 from the eruption, both decreasing from about 1.0 Tg to 0.6 Tg (Fig. 12; NASA/GSFC, 2019, last access: 2020-08-18, https://so2.gsfc.nasa.gov/pix/special/2019/raikoke/raikoke_tropomi_so2stl_0621-07062019.html)

Despite this early underestimation and the fact that the $SO_2$ injection more or less instantaneous compared with the time span of cloud detectability, CrIS total mass (and to a lesser extent, preliminary TROPOMI total mass as well) did not begin the anticipated exponential decay until approximately 15-20 days after $SO_2$ injection. We posit that this was mainly due to CrIS channel saturation and highly irregular dispersal by extratropical cyclonic winds affecting the total CrIS-measureable mass and cloud dilution respectively. However, the presence of this effect in preliminary TROPOMI suggests that at least some contribution may have derived from time-dependent $SO_2$ to sulfate conversion kinetics. As has been invoked in previous studies, this could have been the result of limited hydroxyl radical (OH) early in the cloud history (e.g., Theys et al., 2013; Sekiya et al., 2016), although detailed chemical modelling to support this is beyond the scope of this work. As the e-folding times shown in Fig. 12c,d are calculated by finite difference, they are quite noisy; however, they exhibit the same trend of slow decay early in the cloud history even after CrIS and TROPOMI masses become similar. After about 35 days, the apparent e-

folding for the total VCD settles to a more or less constant value ($\sim$10 day e-folding time). The apparent stratospheric e-folding time exhibits a very different pattern, remaining approximately constant ($\sim$10 day e-folding time) until about day 30, when it begins to shorten. The large uncertainty in the stratospheric e-folding time after approximately 30 days is the result of the increasing relative stratospheric mass uncertainty and the particular details of the PDF calculation (Hinkley, 1969). Because $SO_2$ is typically assumed to have a long lifetime in the stratosphere, this is likely the result of dilution below the detection threshold over large low-concentration regions of the cloud, although some portion of the $SO_2$ re-entering the troposphere as it extends to lower latitudes (where the tropopause is higher) may also play a role considering that the total column does not show this effect as significantly.

## 5   Conclusions

i) New probabilistic enhancement of existing hyperspectral IR $SO_2$ retrieval techniques enable the retrieval of PDFs for $SO_2$ height and partial VCD, providing significant statistical power, precision, and consistency to global $SO_2$ detection, tracking, and characterization efforts. Retrieving these PDFs enables the calculation of many new quantities, including exceedence probabilities for concentration, layer height constraint probabilities, mean concentration profiles, and mass at different layers with uncertainty. Although these capabilities are primarily beneficial for operational $SO_2$ monitoring, these methods are relevant to climatological studies because of the ability to the stratospheric fraction of total mass for a given $SO_2$ cloud if tropopause heights are available from ancillary sources.

ii) This technique is capable of resolving larger VCD values than would be anticipated for a linearized approach due to two factors: (i) the use of the height PDF increases the retrieved total VCD compared with a single height estimate and (ii) the use of a specialized channel subset retrieval that improves the linear approximation when the signal is certain to be dominated by $SO_2$. However, these techniques are still limited in their ability to resolve extreme VCD values (many hundreds of DU) like some of those observed in the recent (2019) eruption of Raikoke Volcano, Kuril Islands. Because of the improved spatial resolution over IASI and the technique's sensitivity, we can resolve heights for small clouds that cannot be resolved well by IASI. Additionally, the technique can adequately resolve height information across a broad range of plume altitudes including the in lower stratosphere, nearly to the surface, though with more limited detection.

iii) Preliminary comparisons suggest that this method generally compares well with other measurements of $SO_2$ VCD and altitude; however, the probabilistic framework adds significant value especially in the retrieval of height information. Cross sections through these probability clouds compare very well with cloud heights from CALIOP lidar backscatter.

iv) As a logical extension of the probabilistic framework, this technique enables the characterization of $SO_2$ clouds for long-term probabilistic analyses of cloud evolution and key time-varying parameters such as total mass, stratospheric mass, and apparent e-folding time.

v) The algorithms presented here are currently being integrated into VOLCAT, where they will be used for operational $SO_2$ cloud detection, characterization, and tracking in support of aviation safety. We anticipate future work to include

more comprehensive comparison of height PDFs with CALIOP lidar backscatter data, application of these techniques to similar instruments including IASI and the Atmospheric Infrared Sounder (AIRS), as well as the development of volcanic degassing and aviation-focused products.

*Code and data availability.* The Level - 1B CrIS data utilized in this study are available from the Goddard Earth Sciences Data and Information Services Center (GES-DISC, https://disc.gsfc.nasa.gov/). The tropopause data used here to define the lower limit of the stratosphere are available from the NOAA Earth Science Research Laboratory Physical Sciences Laboratory (NOAA/OAR/ESRL PSL,https://www.esrl.noaa.gov/psd/data/data.ncep.reanalysis.html) The code developed to generate samples of the $SO_2$-free background spectrum by the NORTA process are available in a git repository at https://gitlab.ssec.wisc.edu/dhyman/trace_gas_background_spectra. This repository also includes a list of the $SO_2$-free days discussed in the text as well as GES-DISC links for all of the CrIS granules collected during those days. The Eumetsat IASI Level-2 $SO_2$ height and VCD data is available from the AERIS IASI portal (https://iasi.aeris-data.fr/). IASI is a joint mission of EUMETSAT and the Centre National dÉtudes Spatiales (CNES, France). The authors acknowledge the AERIS data infrastructure for providing access to the IASI data in this study and ULB-LATMOS for the development of the retrieval algorithms. S5P TROPOMI Level-2 $SO_2$ data is avaiable from the Sentinel-5p Pre-Operations Data Hub https://s5phub.copernicus.eu/dhus/#/home NASA CALIOP data is available from https://www-calipso.larc.nasa.gov.

## Appendix A: Generating Correlated Random Spectra for Monte Carlo Retrieval: NORTA Sampling

The general problem of sampling a correlated random vector ($Y_i \in \boldsymbol{Y} \in \mathbb{R}^N$) with non-normal marginal distributions $F_{Y_i}$ and covariance matrix $\mathbf{S}$ is accomplished by a transform sampling technique known as NORTA (NORmal To Anything) (Cario and Nelson, 1997) in which the desired random vector is written as a component-wise inverse marginal transform of a standard normal random vector:

$$Y_i = F_{Y_i}^{-1}(\Phi(Z_i)) \tag{A1}$$

where $\Phi$ is the univariate standard normal cumulative density function (CDF). In this formulation, the individual marginals can be used to transform the components of the standard normal vector. The crux of this method lies in generating a standard normal random vector $\boldsymbol{Z}$ with appropriate correlation structure $\rho_{\boldsymbol{Z}}$ which upon component-wise transformation as above, will produce the desired random vector $\boldsymbol{Y}$ with the desired covariance structure $\mathbf{S}$. That is, a correlated $\boldsymbol{Z}$ must be generated with the correlation matrix

$$\rho_{\boldsymbol{Z}} = \mathrm{Cov}(\boldsymbol{Z}) = \mathbb{E}\left[\boldsymbol{Z}\boldsymbol{Z}^{\mathsf{T}}\right], \tag{A2}$$

which, after transformation, results in the correlation matrix $\rho_{\boldsymbol{Y}} = \mathrm{Corr}(\boldsymbol{Y})$ associated with the the known covariance matrix $\mathbf{S}$.

This is accomplished by solving

$$\rho_{\boldsymbol{Y}}(i,j) = C_{ij}\left[\rho_{\boldsymbol{Z}}(i,j)\right] \tag{A3}$$

for each of the unique $N(N-1)/2$ correlations ($\rho_{\boldsymbol{Z}}(i,j)$) in the lower triangle of $\rho_{\boldsymbol{Z}}$. The correlation transformation function is

$$C_{ij}\big[\rho_{\boldsymbol{Z}}(i,j)\big] := \frac{\mathbb{E}[Y_i(Z_i)\,Y_j(Z_j)] - \mathbb{E}[Y_i]\,\mathbb{E}[Y_j]}{\sqrt{\mathrm{Var}[Y_i]\,\mathrm{Var}[Y_i]}} \tag{A4}$$

and

$$\mathbb{E}[Y_i(Z_i)\,Y_j(Z_j)] = I\big[\rho_{\boldsymbol{Z}}(i,j)\big] := \iint\limits_{\mathbb{R}^2} F_{Y_i}^{-1}[\Phi(z_i)]\,F_{Y_j}^{-1}[\Phi(z_j)]\,\varphi(z_i,z_j;\rho_{\boldsymbol{Z}}(i,j))\,\mathrm{d}z_i\,\mathrm{d}z_j \tag{A5}$$

where $\varphi(z_i,z_j;\rho_{\boldsymbol{Z}}(i,j))$ is the bivariate standard normal density function with correlation $\rho_{\boldsymbol{Z}}(i,j)$ between the variables $Z_i$ and $Z_j$. These problems may be inverted individually by various methods outlined in the original formulation of Cario and Nelson (1997).

## A1  Application of NORTA to background spectrum

For the purposed of generating the correlated random background spectrum in the present study, we make several modifications to the classical method of Cario and Nelson (1997) to make our problem tractable but retain the high fidelity of the CrIS measurements. In the present study, 177 CrIS channels are used representing the FSR mid-wave band between $1300-1410$ cm$^{-1}$. This yields $15{,}576$ independent correlation - matching inverse problems which are solved in the present study by gradient descent iteration. As in the main text, we first collect a database of SO$_2$-free CrIS spectra for each seasonal $5° \times 5°$ bin and compute the $1300-1410$ cm$^{-1}$ background covariance matrix $\mathbf{S}$ as well as the channel-wise marginal distributions as a sequence of 177 histograms. Then we perform the NORTA process to generate 10,000 samples of $\boldsymbol{Y}_{bg}$ for each season-latitude-longitude bin.

## A2  Numerical Integration of the Joint Expectation

Because each of the correlation - matching inverse problems requires multiple rounds of numerical integration of Eq. A5, we make several modifications to increase computational efficiency. Because many of the channel correlations are strong, the bivariate standard normal distribution $\varphi(z_i,z_j;\rho_{\boldsymbol{Z}}(i,j))$ for each such pair of channels is very narrow. Consequently, for a typical rectangular sampling domain, the integrand of Eq. A5 is approximately zero almost everywhere except for a concentrated region in which accurate numerical integration is challenging. To reduce wasted integrand samples, we make a standard transformation of the bivariate normal distribution and then cast the domain in polar coordinates and approximate the integral over a finite radial domain $[0,R]$:

$$I[\rho_{\boldsymbol{Z}}(i,j)] := \int\limits_{0}^{2\pi}\int\limits_{0}^{R} G_{ij}(r,\theta;\rho_{\boldsymbol{Z}}(i,j))\frac{e^{-r^2/2}}{2\pi}r\,\mathrm{d}r\,\mathrm{d}\theta \tag{A6}$$

where

$$G_{ij}(r,\theta;\rho_{\boldsymbol{Z}}(i,j)) = F_{Y_i}^{-1}[\Phi(r\cos\theta)]\,F_{Y_j}^{-1}[\Phi(\rho_{\boldsymbol{Z}}(i,j)\,r\cos\theta + \sqrt{1-\rho_{\boldsymbol{Z}}^2(i,j)}\,r\sin\theta)] \tag{A7}$$

is the product of the transformed i-th and j-th components of the desired random vector. This approximates the full improper integral within tolerance $\epsilon$, with the fixed value $\epsilon = 10^{-6}$ in the present study. The terminal radius is estimated conservatively as

$$R = \sqrt{2\ln\left(\frac{1}{\epsilon}\frac{\max Y_i \max Y_j}{\min Y_i \min Y_j}\right)} \tag{A8}$$

where the maximum and minimum values of the components were recorded during the generation of $\mathbf{S}$ and the marginal distributions from sample data. In the absence of knowledge of these minima and maxima, they could be estimated from the marginal distributions as fixed percentiles of these distributions.

The estimated radial limit of integration $R$ is derived by requiring that

$$\int_0^{2\pi}\int_R^\infty G_{ij}\frac{e^{-r^2/2}}{2\pi}r\,\mathrm{d}r\,\mathrm{d}\theta \le \epsilon \int_0^{2\pi}\int_0^\infty G_{ij}\frac{e^{-r^2/2}}{2\pi}r\,\mathrm{d}r\,\mathrm{d}\theta. \tag{A9}$$

This inequality can be solved for $R$ under the conservative estimation that $\max G_{ij} = \max Y_i \max Y_j$ and $\min G_{ij} = \min Y_i \min Y_j$ which are derived from the maximum and minimum brightness temperatures on each channel from among all background spectra in the database. This leads to

$$\int_R^\infty re^{-r^2/2}\,\mathrm{d}r \le \epsilon\frac{\min Y_i \min Y_j}{\max Y_i \max Y_j}\int_0^\infty re^{-r^2/2}\,\mathrm{d}r \tag{A10}$$

and upon integrating, $R$ may be chosen conservatively to satisfy

$$e^{-R^2/2} \le \epsilon\frac{\min Y_i \min Y_j}{\max Y_i \max Y_j}. \tag{A11}$$

This transformation to a scaled polar coordinates ensures that the curvature and gradients in the integrand are as small as possible, ensuring that numerical integration is as accurate as possible for a given domain sampling up to the order of the method employed. Additionally, the sensible radial limitation of this integral ensures that it can be computed efficiently to within tolerance without the inclusion of samples which minimally affect the total (approximately Gaussian tails).

## A3 Numerical Solution of Inverse Problems

Although each channel pair inverse problem can be solved separately by Newton's method or other algorithms, we solve all of the problems jointly, restating the problem as a gradient - descent minimization of the total square correlation error

$$\varepsilon^2(\boldsymbol{r_Z}) = \left[\boldsymbol{C}(\boldsymbol{r_Z}) - \boldsymbol{r_Y}\right]^\mathsf{T}\left[\boldsymbol{C}(\boldsymbol{r_Z}) - \boldsymbol{r_Y}\right], \tag{A12}$$

where the vectors $\boldsymbol{r_Z}, \boldsymbol{r_Y} \in \mathbb{R}^{15,576}$ are comprised of the unique lower triangular elements of $\rho_{\boldsymbol{Z}}$ and $\rho_{\boldsymbol{Y}}$ respectively and $\boldsymbol{C}: \mathbb{R}^{15,576} \to \mathbb{R}^{15,576}$ is the correlation transformation function cast as a vector function. Although this seems intractable due to the extreme dimensionality, there is no correlation between these dimensions (no correlation between pairwise inverse problems), and consequently, the Jacobian is zero everywhere except for its main diagonal. The gradient descent method of

Barzilai and Borwein (1988) produces fast convergence to a global minimum due to the monotonicity and bounding properties
of the correlation functions $C_{ij}$ as described by Cario and Nelson (1997). At each iteration, the NORTA processes is completed
and the error on the synthesized channel marginal distributions and spectral covariance matrix are used as convergence criteria.
The vectorization of these independent problems allows for standardization in convergence criteria using global (total) error
minimization. Because the error is measured on the final product (the desired random spectra), a minimum number of iterations
is required by comparison with performing each problem separately and then generating the desired random spectra.

## Appendix B: Derivation of VCD mean, variance, and covariance formulae

As in the text, $SO_2$ is assumed to exist in a narrow, 1 km-thick layer, represented by a box profile:

$$\boldsymbol{x} = X \Pi(h - H) = \begin{cases} X/L & L/2 < h - H < L/2 \\ 0 & \text{otherwise} \end{cases} \tag{B1}$$

where $L$ is 1 km. For the purposes of simplicity in computation, we consider a limiting case of a very thin layer ($L \to 0$), where
the finite box profile converges in distribution to the Dirac delta:

$$\Pi(h - H) \xrightarrow{d} \delta(h - H) \quad \text{as} \quad L \to 0 \tag{B2}$$

The calculation of the mean and variance below make extensive use of Fubini's Theorem allowing reordering of iterated
integration. Because these integrals contain the Dirac delta, it is not simple to show that the conditions of Fubini's theorem are
satisfied due to the Lebesgue non-integrability of the Dirac delta, which in turn stems from the fact that the Dirac delta is not a
proper function, but a distribution. However, we proceed assuming that the iterated integrals can be interchanged. We remark
that a proof using limits of functions ("nascent delta functions") approaching the Dirac delta could be substituted here at great
cost to simplicity and readability. For a finite thickness layer (as assumed in the main text), all convolutions with the Dirac
delta below would be replaced with convolutions with a boxcar function (a finite nascent delta). For the 1 km-thick box profile
above, the results of convolution will be smoothed, 1 km-running-averaged versions of the desired functions. The final integral
formulas would be the same except that the integrands would first be smoothed by a 1 km-running average. In a discrete setting
with 1 km sampling of the retrieved height PDF, the assumption of a 1 km thick box profile yields the same results as using the
Dirac delta.

The mean partial VCD is calculated as the expectation

$$\mathbb{E}[\hat{X}(h)] = \mathbb{E}\left[\int_0^h \hat{X} \delta(\eta - H) \, d\eta\right] = \iint_{\mathcal{H}} \iint_{\mathcal{X}} \left(\int_0^h \hat{x} \delta(\eta - h') \, d\eta\right) f_{\hat{X},H}(\hat{x}, h') \, d\hat{x} \, dh'$$

$$= \iint_{\mathcal{H}} \iint_{\mathcal{X}} \left(\int_0^h \hat{x} \delta(\eta - h') \, d\eta\right) f_{\hat{X}|H}(\hat{x}|h') f_H(h') \, d\hat{x} \, dh' \tag{B3}$$

where $\mathcal{H}$ and $\mathcal{X}$ are the domains of height and VCD (both $\sigma$-finite measure spaces) and the last equality follows from the definition of the conditional PDF. Rearranging this iterated integral gives

$$\mathbb{E}[\hat{X}(h)] = \int_0^h \int_{\mathcal{H}} \delta(\eta - h') f_H(h') \left( \int_{\mathcal{X}} \hat{x} f_{\hat{X}|H}(\hat{x}|h') \, d\hat{x} \right) dh' \, d\eta$$

$$= \int_0^h \int_{\mathcal{H}} \delta(\eta - h') \, f_H(h') \, \mathbb{E}(\hat{X} \,|\, H = h') \, dh' \, d\eta. \quad \text{(B4)}$$

Because of the symmetry of the delta function ($\delta(\eta - h') = \delta(h' - \eta)$), the integral properties of the delta function yields

$$\mathbb{E}[\hat{X}(h)] = \int_0^h f_H(\eta) \, \mathbb{E}(\hat{X} \,|\, H = \eta) \, d\eta. \quad \text{(B5)}$$

Because the algebraic form of the variance is $\mathrm{Var}(\hat{X}(h)) = \mathbb{E}[\hat{X}^2(h)] - [\mathbb{E}(\hat{X}(h))]^2$, only the second moment of the partial VCD $\mathbb{E}[\hat{X}^2(h)]$ must be calculated to complete the formula. This quantity is

$$\mathbb{E}[\hat{X}^2(h)] = \mathbb{E}\left[ \left( \int_0^h \hat{X} \delta(\eta - H) \, d\eta \right)^2 \right]$$

$$= \int_{\mathcal{H}} \int_{\mathcal{X}} \left( \int_0^h \hat{x} \delta(\eta_0 - h') \, d\eta_0 \right) \left( \int_0^h \hat{x} \delta(\eta - h') \, d\eta \right) f_{\hat{X}|H}(\hat{x}|h') f_H(h') \, d\hat{x} \, dh'$$

$$= \int_0^h \int_0^h \int_{\mathcal{H}} \delta(\eta_0 - h') \delta(\eta - h') f_H(h') \left( \int_{\mathcal{X}} \hat{x}^2 f_{\hat{X}|H}(\hat{x}|h') \, d\hat{x} \right) dh' \, d\eta_0 \, d\eta$$

$$= \int_0^h \int_0^h \int_{\mathcal{H}} \delta(h' - \eta_0) \delta(h' - \eta) f_H(h') \, \mathbb{E}(\hat{X}^2 \,|\, H = h') \, dh' \, d\eta_0 \, d\eta$$

$$= \int_0^h \int_0^h \delta(\eta_0 - \eta) f_H(\eta_0) \, \mathbb{E}(\hat{X}^2 \,|\, H = \eta_0) \, d\eta_0 \, d\eta$$

$$\text{(B6)}$$

Since the dummy variable $\eta$ always runs between $0 < \eta < h$, the delta function $\delta(\eta_0 - \eta)$ is always centered in the interval $0 < \eta_0 < h$ and the integral properties of the delta function can be applied again:

$$\mathbb{E}[\hat{X}^2(h)] = \int_0^h f_H(\eta) \, \mathbb{E}(\hat{X}^2 \,|\, H = \eta) \, d\eta. \quad \text{(B7)}$$

Substitution of $\mathbb{E}(\hat{X}^2 \,|\, H = \eta) = \mathrm{Var}(\hat{X} \,|\, H = \eta) + [\mathbb{E}(\hat{X} \,|\, H = \eta)]^2$ into the formula for $\mathbb{E}[\hat{X}^2(h)]$ and subsequently into the formula above for $\mathrm{Var}(\hat{X}(h))$ completes the variance.

Similarly, for the covariance between partial VCD at two altitudes $h = a$ and $h = b$ with $b \geq a$, only the mixed expectation $\mathbb{E}[\hat{X}(b)\hat{X}(a)]$ must be calculated:

$$\mathbb{E}[\hat{X}(a)\hat{X}(b)] = \mathbb{E}\left[ \int_0^a \hat{X}\delta(\eta_a - H)\,\mathrm{d}\eta_a \int_0^b \hat{X}\delta(\eta_b - H)\,\mathrm{d}\eta_b \right]$$

$$= \int_{\mathcal{H}} \int_{\mathcal{X}} \left( \int_0^a \hat{x}\delta(\eta_a - h')\,\mathrm{d}\eta_a \right) \left( \int_0^b \hat{x}\delta(\eta_b - h')\,\mathrm{d}\eta_b \right) f_{\hat{X}\,|\,H}(\hat{x}\,|\,h') f_H(h')\,\mathrm{d}\hat{x}\,\mathrm{d}h'$$

$$= \int_0^a \int_0^b \int_{\mathcal{H}} \delta(h' - \eta_b)\delta(h' - \eta_a) f_H(h')\,\mathbb{E}(\hat{X}^2\,|\,H = h')\,\mathrm{d}h'\,\mathrm{d}\eta_b\,\mathrm{d}\eta_a. \quad \text{(B8)}$$

because the domains $(0, a)$, $(0, b)$, and $\mathcal{H}$ have the relationship

$$(0, a) \subseteq (0, b) \subseteq \mathcal{H}, \tag{B9}$$

it follows that $\eta_a \in \mathcal{H}$, $\eta_b \in \mathcal{H}$, and $0 < \eta_a < \eta_b$, so that

$$\mathbb{E}[\hat{X}(a)\hat{X}(b)] = \int_0^a \int_0^b \delta(\eta_b - \eta_a) f_H(\eta_b)\,\mathbb{E}(\hat{X}^2\,|\,H = \eta_b)\,\mathrm{d}\eta_b\,\mathrm{d}\eta_a$$

$$= \int_0^a f_H(\eta_a)\,\mathbb{E}(\hat{X}^2\,|\,H = \eta_a)\,\mathrm{d}\eta_a = \mathbb{E}[\hat{X}^2(a)]. \quad \text{(B10)}$$

Using the algebraic formula for the variance and covariance, we attain by substitution

$$\mathrm{Cov}[\hat{X}(a), \hat{X}(b)] = \mathrm{Var}[\hat{X}(a)] - \mathbb{E}[\hat{X}(a)]\Big( \mathbb{E}[\hat{X}(b)] - \mathbb{E}[\hat{X}(a)] \Big) \tag{B11}$$

which is always less than $\mathrm{Var}[\hat{X}(a)]$ unless $a = b$ since the partial VCD is cumulative, yielding larger values at higher altitudes.

**Remark:**

Of particular importance, these formulae may be used to calculate the expectation and variance values of the partial VCD between two altitudes. The expected value is

$$\mathbb{E}[\hat{X}(b) - \hat{X}(a)] = \mathbb{E}[\hat{X}(b)] - \mathbb{E}[\hat{X}(a)]. \tag{B12}$$

and the variance is

$$\mathrm{Var}[\hat{X}(b) - \hat{X}(a)] = \mathrm{Var}[\hat{X}(b)] + \mathrm{Var}[\hat{X}(a)] - \mathrm{Cov}[\hat{X}(a), \hat{X}(b)]$$

$$= \mathrm{Var}[\hat{X}(b)] + \mathbb{E}[\hat{X}(a)]\Big( \mathbb{E}[\hat{X}(b)] - \mathbb{E}[\hat{X}(a)] \Big). \quad \text{(B13)}$$

## Appendix C: Bilinear interpolation of background spectrum

To smooth the changes between retrievals in adjacent background cells, we use a bilinear interpolation of the background spectra. For a general quantity $(Q)$, the bilinear interpolation is represented as

$$Q(x,y) = c_x c_y Q(x_0, y_0) + (1 - c_x) c_y Q(x_1, y_0) + c_x(1 - c_y) Q(x_0, y_1) + (1 - c_x)(1 - c_y) Q(x_1, y_1) \tag{C1}$$

between the corner points $(x_0, y_0)$, $(x_1, y_0)$, $(x_0, y_1)$, $(x_1, y_1)$, using the scalings $c_x = (x_1 - x)/(x_1 - x_0)$ and $c_y = (y_1 - y)/(y_1 - y_0)$. We interpolate the inverse error covariance matrix $\mathbf{S}^{-1}$ by this formula. However, because our background spectrum is characterized probabilistically as a set of $N = 10,000$ samples $(\boldsymbol{y}_{bg}^s \in \Omega_{\boldsymbol{Y}_{bg}})$ of a correlated random vector $\boldsymbol{Y}_{bg}$ in each seasonal $5° \times 5°$ background grid cell, the samples cannot be interpolated directly by the above formula. Instead we treat $\boldsymbol{Y}_{bg} = \boldsymbol{Y}_{bg}(X, Y)$ as a function of a random position where $(X, Y)$ is a discrete random position, taking only the cell corner points as possible values. In this case, $X$ and $Y$ represent longitude and latitude. In particular, we characterize $(X, Y)$ by the probability mass function $p(x_i, y_j) = (-1)^{i+j}(i - c_x)(j - c_y)$ for $i, j \in \{0, 1\}$ which is simply the corner point weights in the bilinear interpolation formula. Consequently, we generate $\boldsymbol{Y}_{bg}(x, y) = \mathbb{E}_{(X,Y)}[\boldsymbol{Y}_{bg}(X, Y)] = \sum_{i,j} \boldsymbol{Y}_{bg}(x_i, y_j) p(x_i, y_j)$ by sampling the discrete distribution $p(x_i, y_j)$ to generate the number of samples taken from each of the corner points $n(x_i, y_j) = [Np(x_i, y_j)]$ where the bracket represents rounding to the nearest integer. Using this sampling, we generate the samples of $\boldsymbol{Y}_{bg}(x, y)$ as the collection of each of the $n(x_i, y_j)$ samples from the corners $\boldsymbol{Y}_{bg}(x_i, y_j)$. This generates a total of $N$ samples for the interpolated background spectrum.

## Appendix D: CrIS Channels used in SO2 retrieval

The following CrIS channels are used in this work and are identified by their wavenumber value $(\text{cm}^{-1})$.

### D1 For the regular retrieval:

1300.0 , 1300.625, 1301.25 , 1301.875, 1302.5 , 1303.125, 1303.75 , 1304.375, 1305. , 1305.625, 1306.25 , 1306.875, 1307.5 , 1308.125, 1308.75 , 1309.375, 1310. , 1310.625, 1311.25 , 1311.875, 1312.5 , 1313.125, 1313.75 , 1314.375, 1315. , 1315.625, 1316.25 , 1316.875, 1317.5 , 1318.125, 1318.75 , 1319.375, 1320. , 1320.625, 1321.25 , 1321.875, 1322.5 , 1323.125, 1323.75 , 1324.375, 1325. , 1325.625, 1326.25 , 1326.875, 1327.5 , 1328.125, 1328.75 , 1329.375, 1330. , 1330.625, 1331.25 , 1331.875, 1332.5 , 1333.125, 1333.75 , 1334.375, 1335. , 1335.625, 1336.25 , 1336.875, 1337.5 , 1338.125, 1338.75 , 1339.375, 1340. , 1340.625, 1341.25 , 1341.875, 1342.5 , 1343.125, 1343.75 , 1344.375, 1345. , 1345.625, 1346.25 , 1346.875, 1347.5 , 1348.125, 1348.75 , 1349.375, 1350. , 1350.625, 1351.25 , 1351.875, 1352.5 , 1353.125, 1353.75 , 1354.375, 1355. , 1355.625, 1356.25 , 1356.875, 1357.5 , 1358.125, 1358.75 , 1359.375, 1360. , 1360.625, 1361.25 , 1361.875, 1362.5 , 1363.125, 1363.75 , 1364.375, 1365. , 1365.625, 1366.25 , 1366.875, 1367.5 , 1368.125, 1368.75 , 1369.375, 1370. , 1370.625, 1371.25 , 1371.875, 1372.5 , 1373.125, 1373.75 , 1374.375, 1375. , 1375.625, 1376.25 , 1376.875, 1377.5 , 1378.125, 1378.75 , 1379.375, 1380. , 1380.625, 1381.25 , 1381.875, 1382.5 , 1383.125, 1383.75 , 1384.375, 1385. , 1385.625, 1386.25 , 1386.875, 1387.5 , 1388.125, 1388.75 , 1389.375, 1390. , 1390.625, 1391.25 , 1391.875, 1392.5 , 1393.125, 1393.75 , 1394.375, 1395. , 1395.625,

1396.25 , 1396.875, 1397.5 , 1398.125, 1398.75 , 1399.375, 1400. , 1400.625, 1401.25 , 1401.875, 1402.5 , 1403.125, 1403.75 , 1404.375, 1405. , 1405.625, 1406.25 , 1406.875, 1407.5 , 1408.125, 1408.75 , 1409.375, 1410.0

## D2 For the specialized, high-loading retrieval:

1300. , 1300.625, 1301.25 , 1301.875, 1302.5 , 1303.125, 1303.75 , 1304.375, 1305. , 1305.625, 1306.25 , 1306.875, 1307.5 , 1308.125, 1308.75 , 1309.375, 1310. , 1310.625, 1311.25 , 1311.875, 1312.5 , 1313.125, 1313.75 , 1314.375, 1315. , 1315.625, 1316.25 , 1316.875, 1317.5 , 1318.125, 1318.75 , 1319.375, 1320. , 1320.625, 1321.25 , 1321.875, 1322.5 , 1323.125, 1323.75 , 1324.375, 1325. , 1325.625, 1326.25 , 1326.875, 1327.5 , 1328.125, 1328.75 , 1329.375, 1330. , 1330.625, 1331.25 , 1331.875, 1332.5 , 1362.5 , 1363.125, 1363.75 , 1387.5 , 1388.125, 1388.75 , 1389.375, 1390. , 1390.625, 1391.25 , 1391.875, 1392.5 , 1393.125, 1393.75 , 1394.375, 1395. , 1395.625, 1396.25 , 1396.875, 1397.5 , 1398.125, 1398.75 , 1399.375, 1400. , 1400.625, 1401.25 , 1401.875, 1402.5 , 1403.125, 1403.75 , 1404.375, 1405. , 1405.625, 1406.25 , 1406.875, 1407.5 , 1408.125, 1408.75 , 1409.375, 1410.

## Appendix E: Probabilistic time series

### E1 Probabilistic mass

In general, the total cloud mass can be calculated by integrating the total VCD $\hat{X}(h_\infty)$ over the SO$_2$ cloud region. In the present study, we make this calculation after interpolating the CrIS retrievals onto an equal area grid (grid cells of constant area $\delta A$). Because the set of measurements are normally distributed and assumed independent with means $\mathbb{E}[\hat{X}_i(h_\infty)]$ and variances $\mathrm{Var}[\hat{X}_i(h_\infty)]$, their sum is also normally distributed (DeGroot and Schervish, 2012):

$$M \sim \mathcal{N}(\mathbb{E}(M), \mathrm{Var}(M)) \tag{E1}$$

where the mean is

$$\mathbb{E}(M) = \kappa\, \delta A \sum_i \mathbb{E}[\hat{X}_i(h_\infty)] \tag{E2}$$

and variance is

$$\mathrm{Var}(M) = (\kappa\, \delta A)^2 \sum_i \mathrm{Var}[\hat{X}_i(h_\infty)] \tag{E3}$$

and $M$ has units of kilotons (kt) of SO$_2$ and the factor $\kappa = 2.8617 \times 10^{-11}$ kt m$^{-2}$ DU$^{-1}$ has been included for dimensional consistency with VCD measured in DU. This parameterizes the total cloud mass as a Gaussian PDF for any period of data coverage.

## E2 Probabilistic decay rate coefficient and e-folding time

We treat the above time series of PDFs of $SO_2$ mass as a random process $M_t$. As a continuous process, the conversion of $SO_2$ into sulfur aerosols can be modelled kinetically by the differential equation $\dot{M}_t = -k_t M_t$ where $k_t$ is the instantaneous decay rate coefficient. Below we generate $k_t$ and the e-folding time $\tau_t = k_t^{-1}$ as random processes from $M_t$.

To make this calculation in practice, a finite difference formula is needed for $\dot{M}_t$. We write this as a general $2\alpha$ - order accuracy central finite difference formula for the first derivative:

$$\dot{M}_t \approx \frac{1}{\Delta t} \sum_{i=-\alpha}^{\alpha} \delta_{t+i} M_{t+i} \tag{E4}$$

where $\delta_{t+i}$ are the central difference scheme coefficients. As before, the weighted sum of normal random variables is also normally distributed. Consequently,

$$\dot{M}_t \sim \mathcal{N}\left(\mathbb{E}(\dot{M}_t), \text{Var}(\dot{M}_t)\right) \tag{E5}$$

where the mean is

$$\mathbb{E}(\dot{M}_t) = \frac{1}{\Delta t} \sum_{i=-\alpha}^{\alpha} \delta_{t+i} \mathbb{E}(M_{t+i}) \tag{E6}$$

and variance is

$$\text{Var}(\dot{M}_t) = \frac{1}{(\Delta t)^2} \left[ \sum_{i=-\alpha}^{\alpha} \delta_{t+i}^2 \text{Var}(M_{t+i}) + 2 \sum_{i \neq j} \delta_{t+i} \delta_{t+j} \text{Cov}(M_{t+i}, M_{t+j}) \right] = \frac{1}{(\Delta t)^2} \sum_{i=-\alpha}^{\alpha} \delta_{t+i}^2 \text{Var}(M_{t+i}) \tag{E7}$$

where each covariance is zero because each measurement is independent. Because each $\dot{M}_t$ is normally distributed, this sequence of means and variances fully parameterizes its random process. To calculate $k_t$, we also must calculate $\text{Cov}(M_t, \dot{M}_t)$:

$$\text{Cov}(M_t, \dot{M}_t) = \frac{1}{\Delta t} \mathbb{E}\left[ M_t \sum_{i=-\alpha}^{\alpha} \delta_{t+i} M_{t+i} \right] - \frac{1}{\Delta t} \mathbb{E}(M_t) \sum_{i=-\alpha}^{\alpha} \delta_{t+i} \mathbb{E}(M_{t+i})$$

$$= \frac{1}{\Delta t} \sum_{i=-\alpha}^{\alpha} \delta_{t+i} \text{Cov}(M_t, M_{t+i}) = \frac{1}{\Delta t} \delta_t \text{Var}(M_t) = 0 \tag{E8}$$

where the last two equalities follow from the independence of each $M_t$ and the fact that the central coefficient ($\delta_t$) in any central finite difference for the first derivative is zero. For non-central differences, this is not zero, the last equality does not hold, and $\text{Cov}(M_t, \dot{M}_t) = \frac{1}{\Delta t} \delta_t \text{Var}(M_t)$.

With random processes for the mass and mass rate of change calculated we can calculate the decay rate coefficient as a function of these two random processes:

$$k_t = -\frac{\dot{M}_t}{M_t} \tag{E9}$$

which is a ratio of two Gaussian random processes which may be correlated depending on the finite differencing scheme. The calculation of such a ratio of random variables (Fieller, 1932; Hinkley, 1969) describes the uncertainty of the decay coefficient at each time as a PDF $f_{k_t}(k)$.

Calculating the PDF for the instantaneous e-folding time $\tau_t = k_t^{-1}$ is performed by applying standard rules for functions of random variables (from DeGroot and Schervish, 2012):

$$f_{\tau_t}(\tau) = \frac{1}{\tau^2} f_{k_t}\left(\frac{1}{\tau}\right). \tag{E10}$$

Notably, neither the distributions for decay rate coefficient nor for e-folding time are Gaussian.

*Author contributions.* DMH and MJP conceived of the main concepts. DMH developed the mathematical framework and details for the probabilistic retrieval. DMH developed the code used to construct the background spectra and perform the retrieval. MJP performed all radiative transfer modelling. MJP and DMH analyzed the sensitivity of the Jacobians and developed the specialized retrieval for strong loading. DMH and MJP both tested and tuned the retrieval as well as interpreted and discussed the results. DMH and MJP wrote the manuscript.

*Competing interests.* The authors declare no competing interests.

*Acknowledgements.* The NOAA JPSS Proving Ground and Risk Reduction (PGRR) Program funded this research (Federal Grant number: NA15NES4320001). The authors wish to thank Lieven Clarrise for sharing the data used to construct the $SO_2$-free background timeline as well as all members of the JPSS PGRR Volcanic Hazards Initiative team for feedback on product development. The authors also wish to acknowledge the the helpful comments of two anonymous reviewers. The views, opinions, and findings contained in this report are those of the authors and should not be construed as an official National Oceanic and Atmospheric Administration or U.S. Government position, policy, or decision.

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
