# Peer review of "Probabilistic retrieval of volcanic SO2 layer height and partial column density using the Cross-track Infrared Sounder (CrIS)"

_Atmospheric Measurement Techniques, 2020_

## Referee Comment (RC1) · Anonymous Referee #1 · 3 Jul 2020

The authors present a new technique to retrieve SO2 height from hyperspectral measurements from CrIS satellite instrument. Although the results of this study deserve publication in AMT, the paper leaves a poor impression, essentially because it is not written well. There are numbers of comments that should be addressed before the paper can be accepted for publication. I suggest the author improves the structure of the paper, remove unnecessary equations and terms and follow suggestions below.

Comments:

[Figure]

For a non-expert, the paper is very hard to read and does not follow a logical flow. Overall, the paper is written in a way that is not helping the reader. In many places, the text is unnecessary complicated and not concise enough. There are too many unnecessary equations and terminology. Section 2 is hard to digest, the same information can be better described with less words and equations. The probabilistic approach in particular is the novelty of the paper and should be understandable in a simpler way. It should be clear what SO2 height PDFs represent. Is it somehow a propagation of noise/error on the spectra? Is there atmospheric variability accounted also? How can this be linked to the covariance matrix used? This kind of considerations is not well explained and should be clarified. Consequently the added-value of the proposed approach is not obvious, and it is a pity: what sense has a PDF for a metric such as plume height which is in any case an effective estimate for a complex real SO2 vertical distribution? Without a clearer explanation of what this represents, it is hard to judge. This impression is also reinforced by the absence of estimates/discussion on the systematic sources of uncertainties on the retrievals.

-The CrIS instrument is not introduced. Therefore, it is not clear what CrIS is adding new to existing retrievals (from IASI). Basic information such as overpass time, spatial resolution, instrument performance, etc. would be very helpful. Similarly, a small section is needed on the data products used for comparison with CrIS (including references).

-I think Clarisse et al. (2014) is not using a Dirac delta, but rather a prescribed thickness for the SO2 layer. It is unclear at Eq. 3 point if it is what the author suggests. Later it is written that a 1km thickness is used. I find the text of p4 a bit hard to follow. It could be simpler and avoid introducing formulation and Dirac delta if not strictly needed.

-Section 2.2: What is kernel density estimation? The description is difficult to follow: On line 150, the author writes: "We impose a Gaussian prior with mean and variance given by MC sampling using the model columns that make up the Jacobian with noise added." A Gaussian for what? What noise? All information in one sentence is hard to

digest.

-Section 2.3: at l205, the reader discovers that background spectrum and covariance are interpolated spatially. The way section 2 is written is confusing for non experts and the text should be clearer and simpler.

-Section 2.5: here the covariance matrices and mean spectrum are calculated per season lat-lon boxes, etc. It is understood but how is the probabilistic approach implement for each of this season and box, it is unclear.

-Figure 4 is not needed for understanding the paper. Please consider skipping.

-Nothing is said about system uncertainties despite the fact that strong underestimation is found for Raikoke first overpass. Could volcanic ash produces such strong underestimation?

Section 3:

-l 246: the explanation for the lower SO2 columns due to incomplete coverage is very unlikely. Other instruments like TROPOMI have shown huge columns for an area much larger than the CrIS inter-pixels distance.

-Figure 7d,e: Regarding the red curves, "the high noise S-NPP CrIS FOV 7" has not been introduced before and is impossible to understand for someone not familiar with CrIS. Please clarify or skip.

Section 4: comparison with other data sets are shown but without explanation and references. I suggest to add a small section with a presentation of the data sets with necessary details, e.g. what is the overpass time relative to CrIS, spatial resolution, etc.

-Fig 9: TROPOMI is presented for three heights but it is not mentioned what they represent. The Fig 9 is too qualitative (also because of the log scale used). The least would be to show differences or ratios between the retrievals.

-Please use CALIOP throughout the text. Commuting between CALIOP and CALIPSO is confusing.

-On the bimodal PDF in Fig 10 and discussion on l345-355: this is interesting. The author gives two reasons (real feature/artefact of the method) but is not concluding. I wonder to what extend this bi-modal behavior applies to the full plume (not only over the CALIOP track). If this is a significant feature, it might be possible to know if it is real using forward trajectory calculations and CrIS measurements for the next overpass.

-In eq. 25 ".. of SO2 as a Riemann sum". First this equation is trivial, second, what is "Riemann sum" adding here? This paper will be read by scientists used to scientific notation. The paper is full of these and should be simpler.

-Section 4.2: equation 26 is not needed. On Fig 11, the author infer a total mass of 1 Tg while estimate from the VolRes initiative (based on multiple satellites) is of 1.5 Tg. Please explain the discrepancy. I don't understand what the results on e-folding really brings here, expect speculations.

Conclusions

It is written: "Because of the improved spatial resolution over IASI and the technique's sensitivity, we can resolve small clouds that are undetectable by other means ". I'm not convinced. This sentence is not well supported by the paper. First, the reader cannot judge whether CrIS has a superior spatial resolution than IASI because CrIS has not been presented in the paper. Second, it is unclear that the technique has a superior sensitivity than IASI. What would cause this presumable superior sensitivity ? A better instrument or a better retrieval technique? Please remove or elaborate further.

Typos

-l 26: Becasue -> Because

-l 233: the the -> the

-The use of "FOV" is misleading. FOV means field-of-view. Here it is used as satellite overpass or pixels?

---

## Referee Comment (RC2) · Anonymous Referee #2 · 6 Jul 2020

In this manuscript, Hyman and Pavolonis present a method to derive the altitudes and total column amounts of SO2 distributions caused by volcanic eruptions from infrared nadir sounding instruments. They apply their method to the measurements taken by the Cross-track Infrared Sounder (CrIS). Both retrieved quantities are provided as probability distributions. Especially for height, this is a further development of the method by Clarisse et al. (2014) where one distinct height value is provided. Because of this novelty, I support publication of this contribution in AMT.

However, I do have some points which I strongly suggest to be considered before

publication. While specific comments are listed below, my major concerns are the following:

- The description of the method in chapter 2 is provided in a way which is rather difficult to retrace. Partly this is due to a quite mathematical 'high-level' way with which rather obvious issues are presented. However, some aspects also need a better description with more details provided to understand and be able to reproduce the new method. Thus, I think a major revision of chapter 2 is essential.

- The retrieval of total column amounts of SO2 in the presented method is achieved under the assumption of linearity. However, most previous methods apply for this purpose a non-linear iterative retrieval. E.g. Clarisse et al. (2014) state with regard to their linear retrieved column amounts (a side-product of their SO2-layer-height retrieval): 'The conditions of the retrieval, namely constant Jacobians K and linearity are usually not satisfied. The quantity is therefore an apparent column which should be interpreted as a qualitative estimate of the column.' So the authors have to make clear, if they see their linear retrieval of total column amounts in the same way as more qualitative description (but in that case providing a probability distribution would not make much sense), or provide compelling evidence why an iterative approach should not be necessary here.

Specific comments:

L7 'These methods leverage the relative simplicity of infrared radiative transfer calculations':

It is not at all clear what this should mean? Radiative transfer simulations are no retrieval.

L14 'readily incorporated into Monte Carlo forecasting of volcanic emission transport':

This issue is only mentioned in the Abstract and not discussed in the main text. It should either be skipped or discussed in more detail.

L15 'We highlight results including successes and challenges':

There is no information contained in this statement -> skip

L42 'These methods leverage the relative simplicity of infrared radiative transfer calculations':

See comment regarding abstract L7 above.

L57 'We highlight results including successes and challenges':

See comment regarding abstract L15 above.

Between L61 and L63:

A description of the CrIS instrument is missing. Especially it should be stated why only the $v\_3$ band of SO2 has been used and not also the $v\_1$ band which should be more appropriate for sounding higher amounts of SO2 and at lower altitude since it is much less influenced by water vapour (e.g. see Carboni et al., 2012 for IASI).

L64 '2.1 Classical methods for height retrieval':

There is no consistent description of the classical methods in this paragraph. It is rather confusing since the descriptions of 'classical' methods are intermixed with the new method presented. I would strongly suggest to concentrate here on the previous 2-3 methods. I would suggest to substantiate this description by providing a table about the characteristics of these and adding in this table the characteristics of the new approach. This would make it much easier for the reader to get an overview of the major similarities and differences.

Further, since height is a major retrieval quantity coped with in this manuscript, a physical explanation should be provided why there is any information on SO2 plume height in nadir spectra. (According to Clarisse et al., 2014 this is mainly due to the interference with water vapor lines.)

L68 'by Carboni et al. (2012), Clarisse et al. (2014), and Carboni et al. (2016) utilized the ability to linearize a forward radiative transfer model around a climatological mean

state for the concentration of trace SO2,..':

But, Carboni et al., perform non-linear retrievals, as well as Clarisse et al., 2014 in case of column amounts.

L72 'As in Pavolonis (2010), all radiative transfer model simulations used in this study …':

This is not fitting in this paragraph (see comment on L 64 above).

L88 'retrieving only the total column SO2, the mean pressure and the standard deviation (spread).':

Why 'only'? There is hardly more in formation in nadir-observations of SO2. Further, Carboni et al., 2012 mention that in case of low and mid-sized eruptions, the spread is not retrieved.

L95 'concentration profile':

What is the unit of the 'concentration profile' (concentration vs. column); please use 'partial column' to describe clearly the variable.

L99 'Below we refer to the height-dependent Jacobian calculated at the zero-background':

Nowhere in the manuscript is it clearly stated which input parameters are used to determine the Jacobians. E.g. which atmospheric conditions (pressure, temperature, water vapour concentration profiles) have been used for their calculation (meteorological analyses?, observations from CrIS?)?

L117 'it is unsuitable for the height retrieval in particular as follows. Instead of calculating..':

After ending of the first sentence I would have expected an explanation why this is unsuitable. However, there follows the description how it is implemented in the new

method.

L118 'Y':

Please describe more clearly what Y contains. Why is it called a vector, but it is a matrix. Please provide the dimensions of all the vectors/matrices used here. (AMT rule: 'Matrices are printed in boldface, and vectors in boldface italics.')

L118 'correlated':

Please describe which dimension is correlated and why (spectrally?, temporally?).

L122, Eq. 8:

What is the difference here compared to S used by Clarisse et al., 2014?

L123 'samples':

Please state clearly which samples are meant.

L 125, Eq. 9 '(y-Y)':

What is meant here? Is it a difference between the actual spectrum 'y' to each of the background spectra contained in 'Y' ? But one cannot construct an 'S' for each of those single spectra.

L131 'because the z-score is the sum over all of the channels in bvecY':

(1) I don't think it is only the 'sum' of all channels. (2) What is bvecY?

L142 'The likelihood function is constructed by Monte-Carlo (MC) sampling of Y and retrieving the height due to a background spectrum given by MC random sampling according to the marginal PDFs of Y and its covariance S (Fig. 1). The process for sampling this non-Gaussian correlated random vector is detailed in Appendix A.':

This should be explained more in detail and more clearly. What dimensions do the vectors have? Do you use any artificially noisy spectra? How/why do you have to

normalize the samples like in Appendix A.? There should be more description here.

L150 'We impose a Gaussian prior with mean and variance given by MC sampling using the model columns that make up the Jacobian with noise added.':

This description has to be expanded. What is the relation of the MC sampling to the MC sampling when calculation the Likelihood just before?

L151 'specifically, we use the Jacobian corresponding to the traditional Clarisse et al. (2014) height retrieval (h_C).':

How are those Jacobians calculated here? Where do p,T, H2O-concentration profiles come from?

L158, Eq. 15:

In this Bayesian formulation it should be made clearer what is the proposition A and what the evidence B. (E.g. if 'A' means 'SO2 layer at altitude h' and 'B' means 'actual spectrum y measured', then the Likelihood L should be B|A, i.e. 'actual spectrum y measured'|'SO2 layer at altitude h'. I don't see that Eq. 13 defines L like this – please make this clearer in the text.

L164 'Probabilistic Mass Loading':

Above I've expressed my concern about not applying an iterative retrieval here.

L230 'partitioning the data into four seasons':

As I've understood, spatially a smoothness is achieved by interpolation. However, how are jumps avoided due to this seasonal sampling?

L235:

A section about the influence of other effects on the error budget is missing. How, e.g. does ash in combination with SO2 affect the retrieval results? (Since ash is not included in the background S-matrix).

L306:

Can you weight the different explanations for not detecting the southern SO2-cloud? Given Fig. 9f, it seems clear that the initial z-score is more important.

L311 'with a z-score below this threshold':

Why is a z-score threshold at all used in this method? Does this example not show that at least a lower threshold would provide more information?

L311 'this discrepancy may also be due to spectral interference from water vapor in the CrIS SO2 infrared band':

Is there any indication that the water vapor vertical distribution is strongly different from the one further north?

L320 'although exact comparison is not possible due the orbital separation between the satellites carrying IASI (METOP-A,B) and those carrying CrIS':

But you could apply the IASI cloud-height method to your CrIS dataset and compare both. It would be very interesting to see the difference, e.g. as a further plot in Fig. 9 and Fig. 8.

L342 'are first interpolated to fill space and then sampled at the points given by the CALIPSO':

What is the difference with directly interpolating to CALIPSO?

L343 'there is good agreement':

'good agreement' does contain no information. Please try to be more quantitative.

L346 'leading to unrealistically high altitudes there (Fig. 10 e).':

I've tried to detect those in the Figure, but this is very difficult. Please be more specific.

L353 'If the background spectrum has multiple modes (for example, one mode representing deep convective cloud radiances and another for cloud-free radiances), then multiple populations of the Monte-Carlo height samples may accumulate, leading to a multimodal height PDF':

This should be discussed in the section about systematic error sources (see comment to L235).

L374-390:

In my opinion, the discussion about the e-folding-times is out of scope of the actual paper. Therefore, it would be better to skip it. (E.g. the authors do not discuss, that this is an apparent time because the effect of dilution and therefore not being detectable any more for the nadir-sounder is also included in this measure. )

L391 'Conclusions':

Please discuss also the limitations of the method applied to CrIS: no retrievals in nue_1 band of SO2, only linear assumption for retrieval of total column amounts.

L402 'improved spatial resolution over IASI':

Please mention here explicitly the CrIS and IASI pixel-size (km x km). What is the S/N in comparison of the two instruments?

L414 'analysis of errors in the trace gas technique induced by a warming background atmosphere':

What does this mean? Please provide an example.

Technical comments:

L24 'subtly':

-> 'subtle.

L27 'Because':

-> 'Because'

L308 '(f)':

-> '(d)' (first appearance)

L395 'exceedence':

-> 'exceedance'

Fig. 11:

The difference between 'red' and 'blue', and (a) and (b) are not clear.

L412 'applyication':

-> 'application'

L427 'CDF':

Please spell out.

References:

Carboni, E., Grainger, R., Walker, J., Dudhia, A., and Siddans, R.: A new scheme for sulphur dioxide retrieval from IASI measurements: Application to the Eyjafjalla-jökull eruption of April and May 2010, Atmos. Chem. Phys., 12, 11417–11434, doi:10.5194/acp-12-11417-2012, 2012.

Clarisse, L., Coheur, P.-F., Theys, N., Hurtmans, D., and Clerbaux, C.: The 2011 Nabro eruption, a SO2 plume height analysis using IASI measurements, Atmos. Chem. Phys., 14, 3095–3111, doi:10.5194/acp-14-3095-2014, 2014.

---

## Author Comment (AC1) · 25 Aug 2020

Dear Referee #1, attached you can find a detailed letter of response to your comments on the manuscript, as well as to the comments of the other referee. The document includes the revised manuscript, according to the letter. Our responses are in red text delimited by <>.

Best wishes,

Dave Hyman, on behalf of the authors.

Please also note the supplement to this comment:
https://amt.copernicus.org/preprints/amt-2020-41/amt-2020-41-AC1-supplement.pdf

**Supplement:**

Review Response:

Probabilistic retrieval of volcanic SO2 layer height and cumulative mass loading using the Cross-track Infrared Sounder (CrIS)

AMT-2020-41
* * *
Referee 1 (RC1):

The authors present a new technique to retrieve SO2 height from hyperspectral mea-surements from CrIS satellite instrument. Although the results of this study deserve publication in AMT, the paper leaves a poor impression, essentially because it is not written well. There are numbers of comments that should be addressed before the paper can be accepted for publication. I suggest the author improves the structure of the paper, remove unnecessary equations and terms and follow suggestions below.

**<ok>**

**Comments:**

For a non-expert, the paper is very hard to read and does not follow a logical flow.Overall, the paper is written in a way that is not helping the reader. In many places, the text is unnecessary complicated and not concise enough. There are too many unnecessary equations and terminology. Section 2 is hard to digest, the same information can be better described with less words and equations.

<We have revised Section 2 for clarity and simplicity. There are now fewer equations, keeping only the absolutely most essential. Although we feel the the formalism of the probabilistic approach requires the precision of deliberate

definitions and formulae, we accede to this request and have described some elements in text, and have moved some things to the appendices.>

The probabilistic approach in particular is the novelty of the paper and should be understandable in a simpler way. It should be clear what SO2 height PDFs represent. Is it somehow a propagation of noise/error on the spectra? Is there atmospheric variability accounted also? How can this be linked to the covariance matrix used? This kind of considerations is not well explained and should be clarified. Consequently the added-value of the proposed approach is not obvious, and it is a pity: what sense has a PDF for a metric such as plume height which is in any case an effective estimate for a complex real SO2 vertical distribution? Without a clearer explanation of what this represents, it is hard to judge.

< We have made several efforts throughout to better explain what the height PDFs represent. Formally, they represent the probability of finding the detected SO2 layer at a given height. They are a representation of the uncertainty propagated from the differences between the true SO2-free background spectrum and the climatological average SO2-free background as is used in the trace gas method of Clarisse et al, 2014. This is done by MC sampling of many possible background spectra with the sampling constrained by the covariance matrix (S) and the set of channel-wise marginal distributions (histograms) that were carefully collected from a database of over 300 million SO2-free CrIS spectra. The spectral variability included in these samples includes all of the physically realistic conditions that the SO2-free background might have including the effects of differences in the water vapor and temperature profiles, the presence or absence of cloud layers, etc. >

This impression is also reinforced by the absence of estimates/discussion on the systematic sources of uncertainties on the retrievals.

<We have made it much more clear in the revision that the principal source of error in the approach is derived from the linearization of the forward model in solving the inverse problem (channel saturation) and that the technique is really just an accounting of how uncertainty in the background spectrum is propagated through the Clarisse et al., 2014 layer height retrieval. There are other sources as well, such as interference with water vapor, obstruction by high level opaque clouds and instrument noise, but these are not easily quantified and based on our results, it is clear that the algorithm performs pretty well and that with the exception of the channel saturation, these auxiliary sources of error are manageable>

-The CrIS instrument is not introduced. Therefore, it is not clear what CrIS is adding new to existing retrievals (from IASI). Basic information such as overpass time, spatial resolution, instrument performance, etc. would be very helpful. Similarly, a small section is needed on the data products used for comparison with CrIS (including references).

<We have included a section introducing the characteristics of CrIS in the introduction. We have also added a brief description of the comparison products where relevant.>

-I think Clarisse et al. (2014) is not using a Dirac delta, but rather a prescribed thickness for the SO2 layer. It is unclear at Eq. 3 point if it is what the author suggests. Later it is written that a 1km thickness is used. I find the text of p4 a bit hard to follow. It could be simpler and avoid introducing formulation and Dirac delta if not strictly needed.

<We understand that this was a point of confusion and have moved all invocations of the Dirac delta to the appendices. We do use a 1 km-thick box profile in modelling and in our general theoretical framework, but we make a limiting assumption to the Dirac delta to derive the equations for the partial VCD as explained in the appendices.>

-Section 2.2: What is kernel density estimation?

<KDE is a standard technique to estimate the PDF corresponding to a set of samples, similar to a histogram. We have included a standard reference to KDE methods.>

The description is difficult to follow:On line 150, the author writes: "We impose a Gaussian prior with mean and variance given by MC sampling using the model columns that make up the Jacobian with noise added." A Gaussian for what? What noise? All information in one sentence is hard to digest.

<Thank you for pointing this out. We have clarified our presentation of the Bayesian technique for height PDF construction. The prior is essentially the regular Clarisse et al. (2104) retrieval with uncertainty determined in a very similar way to Fig. 2 in that paper. >

-Section 2.3: at I205, the reader discovers that background spectrum and covariance are interpolated spatially. The way section 2 is written is confusing for non experts and the text should be clearer and simpler.

<We have made significant changes to Section 2 to improve readability by a general audience.>

-Section 2.5: here the covariance matrices and mean spectrum are calculated per season lat-lon boxes, etc. It is understood but how is the probabilistic approach implement for each of this season and box, it is unclear.

<We have clarified throughout that the probabilistic approach is applied broadly, with the underlying sources of uncertainty characterized on each season-lat-lon bin. There is a set of 10,000 possible background spectra for each bin. The mean spectrum and (inverse of) the covariance matrix can be interpolated at each lat-lon in the typical way, but a specialized interpolation technique is required to sample the possible backgrounds. It consists of sampling the different sets of background sample spectra from the four nearest lat-lon bins in a specialized way to create a new sample of 10000 background spectra for the queried lat-lon. This process is detailed in the appendix.>

-Figure 4 is not needed for understanding the paper. Please consider skipping.

<In the revision, we have clarified the probabilistic approach, in which Fig. 4 is now a clarifying example of the probabilistic background spectrum.>

-Nothing is said about system uncertainties despite the fact that strong underestimation is found for Raikoke first overpass.

< We provide many new details in the revision regarding this underestimation, including a new figure directly comparing CrIS to TROPOMI in two regions of the Raikoke cloud. This makes it clear that the underestimation is the result of the linearized approach failing under these extreme SO2 loadings. Where more moderate concentrations predominate, CrIS and TROPOMI are in much better agreement, which is the regime that this method is designed to target.>

Could volcanic ash produces such strong underestimation? <Based on the evidence detailed in Carboni et al., 2012, this is very unlikely in the SO2 nu\_3 band where this analysis occurs. This is included in the revision.>

Section 3:

-I 246: the explanation for the lower SO2 columns due to incomplete coverage is very unlikely. Other instruments like TROPOMI have shown huge columns for an area much larger than the CrIS inter-pixels distance.

<Thank you for pointing this out. Although true that TROPOMI showed very large columns over a large area, it is not uniform and maxima are isolated pixels or pixel clusters. We have included a new figure directly comparing CrIS and TROPOMI for the Raikoke cloud that addresses some of these issues. There certainly are sizeable gaps in the coverage of CrIS (approximately 29%-32% by our own estimate) and the larger CrIS footprint compared with TROPOMI also produces some apparent dilution by spatial averaging. This is clear simply by examining histograms of TROPOMI at native resolution vs TROPOMI averaged to CrIS footprints which is included in the new figure. The underestimate is derived from multiple contributing factors including channel saturation (linearization errors), spatial gaps, and the increased footprint size compared to TROPOMI. These arguments are included in the revision.>

-Figure 7d,e: Regarding the red curves, "the high noise S-NPP CrIS FOV 7" has not been introduced before and is impossible to understand for someone not familiar withCrIS. Please clarify or skip.

<As requested, we have included an introduction to the CrIS instrument, which explains the high noise S-NPP CrIS FOV 7. We have left this information in since it is relevant to observe the effect of increased sensor noise on the height PDF retrieval.>

Section 4: comparison with other data sets are shown but without explanation and references. I suggest to add a small section with a presentation of the data sets with necessary details, e.g. what is the overpass time relative to CrIS, spatial resolution, etc.

**<Thank you for this comment we have included a very brief intro to these data.>**

-Fig 9: TROPOMI is presented for three heights but it is not mentioned what they represent. The Fig 9 is too qualitative (also because of the log scale used). The least would be to show differences or ratios between the retrievals.

<As mentioned, we have included more context for this figure and added another that is a direct comparison between CrIS and TROPOMI. We have kept the log- color scale since it is the most appropriate scaling for comparing these data across the many orders of magnitude spanned in such a large cloud. If a linear scale were used, only the regions of highest VCD could be compared, sacrificing the possibility of any broad comparisons of the low-moderate concentration regions of the cloud>

-Please use CALIOP throughout the text. Commuting between CALIOP and CALIPSO is confusing.

**<Ok, this has been changed>**

-On the bimodal PDF in Fig 10 and discussion on I345-355: this is interesting. Theauthor gives two reasons (real feature/artefact of the method) but is not concluding. Iwonder to what extent this bi-modal behavior applies to the full plume (not only over the CALIOP track). If this is a significant feature, it might be possible to know if it is real using forward trajectory calculations and CrIS measurements for the next overpass.

<We have included a bit more on this interesting observation, including that preliminary investigation suggests that such occurrences are not exceedingly rare, but also far from universal. We agree that these observations deserve more detailed study including trajectory calculations; however, thorough investigation of these are beyond the scope of this work. Consequently, we have used more precise language in the revision to limit speculation on these features and their significance and mention them mainly to highlight the diversity of height PDFs that are enabled by this non-parametric technique.>

-In eq. 25 ".. of SO2 as a Riemann sum". First this equation is trivial, second, what is "Riemann sum" adding here? This paper will be read by scientists used to scientific notation. The paper is full of these and should be simpler.

<Thank you for raising this, we have moved these formal points to the appendix where they are necessary for explaining the chain of uncertainty propagation to the probabilistic time series.>

-Section 4.2: equation 26 is not needed. On Fig 11, the author infer a total mass of 1Tg while estimate from the VolRes initiative (based on multiple satellites) is of 1.5 Tg.Please explain the discrepancy.

<Thank you for raising this. We have included a new figure and discussion dealing with this discrepancy in detail.>

I don't understand what the results on e-folding really brings here, expect speculations.

<We have made it clear in the revision that this is an "apparent" e-folding time encompassing all sources of SO2 loss including chemistry as well as dilution and detection threshold-induced loss. We feel that this is a valuable analysis, especially after several weeks since it is (to our knowledge) the first of its kind for the Raikoke eruption cloud and that probabilistic time series are a logical extension of the uncertainty propagation at the core of this work.>

**Conclusions**

It is written: "Because of the improved spatial resolution over IASI and the technique's sensitivity, we can resolve small clouds that are undetectable by other means ". I'm not convinced. This sentence is not well supported by the paper. First, the reader cannot judge whether CrIS has a superior spatial resolution than IASI because CrIS has not been presented in the paper. Second, it is unclear that the technique has a superior sensitivity than IASI. What would cause this presumable superior sensitivity? A better instrument or a better retrieval technique? Please remove or elaborate further.

<We have included a section introducing CrIS which supports these claims about resolution. The claim that CrIS can detect smaller, more dilute clouds is supported by the fact that the early Bogoslof eruption was not detected by IASI. This point has been included in the revision.> Typos -I 26: Becasue -> Because -I 233: the the -> the

<We have fixed these and made every effort to avoid these in the revision.>
* * *
Referee 2 (RC2):

In this manuscript, Hyman and Pavolonis present a method to derive the altitudes and total column amounts of SO2 distributions caused by volcanic eruptions from infrared nadir sounding instruments. They apply their method to the measurements taken by the Cross-track Infrared Sounder (CrIS). Both retrieved quantities are provided as probability distributions. Especially for height, this is a further development of the method by Clarisse et al. (2014) where one distinct height value is provided. Because of this novelty, I support publication of this contribution in AMT.

<Thank you.>

However, I do have some points which I strongly suggest to be considered before publication. While specific comments are listed below, my major concerns are the following:

- The description of the method in chapter 2 is provided in a way which is rather difficult to retrace. Partly this is due to a quite mathematical 'high-level' way with which rather obvious issues are presented. However, some aspects also need a better description with more details provided to understand and be able to reproduce the new method. Thus, I think a major revision of chapter 2 is essential.

<We have made a substantial revision to Section 2, excising many formalisms and adding context for those that remain. The revised Section 2 is much more readable by a general audience and all derivations have been moved to the appendix.>

- The retrieval of total column amounts of SO2 in the presented method is achieved under the assumption of linearity. However, most previous methods apply for this purpose a non-linear iterative retrieval. E.g. Clarisse et al. (2014) state with regard to their linear retrieved column amounts (a side-product of their SO2-layer-height retrieval): 'The conditions of the retrieval, namely constant Jacobians K and linearity are usually not satisfied. The quantity is therefore an apparent column which should be interpreted as a qualitative estimate of the column.' So the authors have to make clear, if they see their linear retrieval of total column amounts in the same way as more qualitative description (but in that case providing a probability distribution would not make much sense), or provide compelling evidence why an iterative approach should not be necessary here.

<Thank you for raising this issue, we have clarified our intention in the revision and have further clarified the effectiveness (and problems) of the linearized approach for vertical column density (VCD). We agree with Clarisse et al., 2014 that VCD is a side-product of height, but the point here is to demonstrate how using a height PDF as opposed to a single height estimate changes how VCD would be calculated. Though true that an actually linear forward model is not typically satisfied, the specialized retrieval for strong loading helps extend the radius of convergence of the linear approximation in cases where a strong SO2 signal is assured. We have included a new figure directly comparing our VCDs with those from TROPOMI (which uses a UV DOAS technique). The new figure highlights the moderate-good accuracy of this CrIS technique for low-moderate VCD and performs with an ~25% underestimate for the extremely large VCD values found in the Raikoke cloud.> Specific comments:

L7 'These methods leverage the relative simplicity of infrared radiative transfer calculations': It is not at all clear what this should mean? Radiative transfer simulations are no retrieval.

**<We have removed this because it is confusing.>**

L14 'readily incorporated into Monte Carlo forecasting of volcanic emission transport': This issue is only mentioned in the Abstract and not discussed in the main text. It should either be skipped or discussed in more detail.

**<We have removed this.>**

L15 'We highlight results including successes and challenges': There is no information contained in this statement -> skip

**<We have removed this.>**

L42 'These methods leverage the relative simplicity of infrared radiative transfer calcu-lations':See comment regarding abstract L7 above.

**<We have removed this.>**

L57 'We highlight results including successes and challenges':See comment regarding abstract L15 above.

**<We have removed this.>**

Between L61 and L63: A description of the CrIS instrument is missing. Especially it should be stated why only the  $v_3$  band of SO2 has been used and not also the  $v_1$  band which should be more appropriate for sounding

higher amounts of SO2 and at lower altitude since it is much less influenced by water vapour (e.g. see Carboni et al., 2012 for IASI).

**<We have included an introduction to the CrIS instrument including consideration regarding the nu1, nu3 bands (CrIS covers only the nu3 band).>**

L64 '2.1 Classical methods for height retrieval': There is no consistent description of the classical methods in this paragraph. It is rather confusing since the descriptions of 'classical' methods are intermixed with the new method presented. I would strongly suggest to concentrate here on the previous 2-3 methods. I would suggest to substantiate this description by providing a table about the characteristics of these and adding in this table the characteristics of the new approach. This would make it much easier for the reader to get an overview of the major similarities and differences.

<Thank you for this suggestion. We have now included a table summarizing the three main methods: Carboni et al 2012, Clarisse et al., 2014, and our CrIS method>

Further, since height is a major retrieval quantity coped with in this manuscript, a physical explanation should be provided why there is any information on SO2 plume height in nadir spectra. (According to Clarisse et al., 2014 this is mainly due to the interference with water vapor lines.)

**<Thank you for this suggestion, we have included it.>**

L68 'by Carboni et al. (2012), Clarisse et al. (2014), and Carboni et al. (2016) utilized the ability to linearize a forward radiative transfer model around a climatological mean state for the concentration of trace SO2,..': But, Carboni et al., perform non-linear retrievals, as well as Clarisse et al., 2014 in case of column amounts.

<Thank you for this correction, it is corrected in the revision.>

L72 'As in Pavolonis (2010), all radiative transfer model simulations used in this study...':This is not fitting in this paragraph (see comment on L 64 above).

<In our revision of Section 2 this now appears with other information about the forward model where we discuss the Jacobians.>

L88 'retrieving only the total column SO2, the mean pressure and the standard devia-tion (spread).': Why 'only'? There is hardly more in formation in nadir-observations of SO2. Further, Carboni et al., 2012 mention that in case of low and mid-sized eruptions, the spread is not retrieved.

**<This is revised, and detailed in the new table.>**

L95 'concentration profile': What is the unit of the 'concentration profile' (concentration vs. column); please use 'partial column' to describe clearly the variable.

<Thank you for raising this terminological point. Throughout the revision, we have used more consistent terminology to distinguish between the total column (``total VCD"), partial column (``partial VCD"), conditional column (``conditional VCD", total column if all SO2 is at height h), and concentration profile. In this instance "concentration profile" was correct since the units were DU km^-1 (mass / volume) and it was a function of height, though the formal definition of the assumed profile is now only in the appendix.>

L99 'Below we refer to the height-dependent Jacobian calculated at the zero-background': Nowhere in the manuscript is it clearly stated which input parameters are used to determine the Jacobians. E.g. which atmospheric conditions (pressure, temperature,water vapour concentration profiles) have been used for their calculation (meteorological analyses?, observations from CrIS?)?

<We have included this information in the revision.>

L117 'it is unsuitable for the height retrieval in particular as follows. Instead of calculating..': After ending of the first sentence I would have expected an explanation why this is unsuitable. However, there follows the description how it is implemented in the new method.

<We have clarified this, though it is a nuanced point which requires some exposition.>

L118 'Y': Please describe more clearly what Y contains. Why is it called a vector, but it is amatrix. Please provide the dimensions of all the vectors/matrices used here. (AMTrule: 'Matrices are printed in boldface, and vectors in boldface italics.')

<The revision uses a slightly different notation to make these quantities clear. Also we have conformed to AMT's style guide in the revision. The bold, roman, capitalized Y (now bold, italicized, capital Y\_bg) was a random vector which is properly a vector in which each element (a spectral channel) is a random variable. Computationally, we work with a collection of realizations of this random variable arranged into an array; however, they are really just samples of this vector, not a matrix. In general, deterministic quantities or realizations of random variables are lower case and the random variables themselves are capitalized (the forward model, Jacobian and covariance matrix are exceptions to this capitalization rule). Samples of random variables are lower case with a superscript s.>

L118 'correlated': Please describe which dimension is correlated and why (spectrally?, temporally?).

<This is spectral correlation, that is, the covariance matrix gives the correlation between elements of the random vector.>

L122, Eq. 8: What is the difference here compared to S used by Clarisse et al., 2014?

<it is constructed in the same way, detailed by Walker et al., 2011.>

L123 'samples': Please state clearly which samples are meant.

<We have done so.>

L 125, Eq. 9 '(y-Y)': What is meant here? Is it a difference between the actual spectrum 'y' to each of the background spectra contained in 'Y'? But one cannot construct an 'S' for each of those single spectra.

<All equations that contain random variables are a theoretical construct. Equations involving samples or means are real in the sense that they are what is computed. This is made much clearer in the new notation.>

L131 'because the z-score is the sum over all of the channels in bvecY': (1) I don't think it is only the 'sum' of all channels. (2) What is bvecY?

<It is true that it is not exactly the sum, it is a weighted sum, but that is not generally so important for purposes of convergence in central limit theorems. "bvecY" was a Latex typo, we regret the mistake.>

L142 'The likelihood function is constructed by Monte-Carlo (MC) sampling of Y and retrieving the height due to a background spectrum given by MC random sampling according to the marginal PDFs of Y and its covariance S (Fig. 1). The process for sampling this non-Gaussian correlated random vector is detailed in Appendix A.': This should be explained more in detail and more clearly. What dimensions do the vectors have? Do you use any artificially noisy spectra? How/why do you have to normalize the samples like in Appendix A.? There should be more description here.

<The revision is much more clear about these issues>

L150 'We impose a Gaussian prior with mean and variance given by MC sampling using the model columns that make up the Jacobian with noise added.':This description has to be expanded. What is the relation of the MC sampling to the MC sampling when calculation the Likelihood just before?

<We have expanded the description of the Bayesian approach. The MC sample background spectra are used in the prior only as to construct a convenient source of zero-mean noise.>

L151 'specifically, we use the Jacobian corresponding to the traditional Clarisse et al.(2014) height retrieval (h\_C).': How are those Jacobians calculated here? Where do p,T, H2O-concentration profiles come from?

<This is described in the revision where the Jacobians are defined.>

L158, Eq. 15: In this Bayesian formulation it should be made clearer what is the proposition A and what the evidence B. (E.g. if 'A' means 'SO2 layer at altitude h' and 'B' means 'actual spectrum y measured', then the Likelihood L should be B|A, i.e. 'actual spectrum y measured'|'SO2 layer at altitude h'. I don't see that Eq. 13 defines L like this – please make this clearer in the text.

<We have clarified the logic of this section. I believe the confusion here was just a matter of notation.>

L164 'Probabilistic Mass Loading': Above I've expressed my concern about not applying an iterative retrieval here.

<The revision is much more clear about the linearization, especially in the section about the specialized strong loading retrieval in which we make the problem approximately linear when we are certain that SO2 is dominating the signal. Our performance against TROPOMI (revised Fig. 9,10) demonstrate that this linearized approach does produce acceptable results.>

L230 'partitioning the data into four seasons': As I've understood, spatially a smoothness is achieved by interpolation. However, how are jumps avoided due to this seasonal sampling?

<We do not deal with temporal interpolation; however, this do not appear to be a problem at this time.>

L235:A section about the influence of other effects on the error budget is missing. How,e.g. does ash in combination with SO2 affect the retrieval results? (Since ash is not included in the background S-matrix)

<Our description of the CrIS instrument discusses this a bit, but as in Carboni et al., 2012, ash interference is minor in the nu\_3 band. That description also discusses interference with water vapor and clouds>

L306: Can you weight the different explanations for not detecting the southern SO2-cloud? Given Fig. 9f, it seems clear that the initial z-score is more important.

<The initial z-score field is the most robust indicator of the presence of SO2; however, it is used mainly as a detection criterion. We have included a better description of these factors.>

L311 'with a z-score below this threshold': Why is a z-score threshold at all used in this method? Does this example not show that at least a lower threshold would provide more information?

<A threshold is necessary to pre-screen out FOVs on which there will be an unreliable full retrieval. The z>5 threshold is used in the previous studies and seems to strike a good balance between catching as much real SO2 as possible and limiting false positives.>

L311 'this discrepancy may also be due to spectral interference from water vapor in the CrIS SO2 infrared band': Is there any indication that the water vapor vertical distribution is strongly different from the one further north?

**<We have included a better description of these factors.>**

L320 'although exact comparison is not possible due the orbital separation between the satellites carrying IASI (METOP-A,B) and those carrying CrIS': But you could apply the IASI cloud-height method to your CrIS dataset and compare both. It would be very interesting to see the difference, e.g. as a further plot in Fig. 9 and Fig. 8.

<We have not done this as the height fields are relatively easy to compare visually on Figure 9 and a thorough comparison is outside the scope of this work. We have included a comparison of CrIS and TROPOMI for VCD, which was necessary based on the other reviewer's point that it is more difficult to compare in Fig 9. Lastly, the IASI data shown probably includes some other steps outside of the exact use of the algorithm of Clarisse et al., 2014, as evidenced by its smoothness, though we were unable to determine what exactly is the difference as only that reference is given with the data for height retrieval.>

L342 'are first interpolated to fill space and then sampled at the points given by the CALIPSO': What is the difference with directly interpolating to CALIPSO?

<This was a confusing sentence. We have clarified that we perform nearest neighbor interpolation to the CALIOP track.>

L343 'there is good agreement': 'good agreement' does contain no information. Please try to be more quantitative.

<As this comparison to CALIOP is clearly qualitative, we feel that "good agreement" reflects the common notion of approximate collocation. In any case, the text after this is more specific.>

L346 'leading to unrealistically high altitudes there (Fig. 10 e).': I've tried to detect those in the Figure, but this is very difficult. Please be more specific.

<We admit this data is hard to see on the map because it is small in extent and directly beneath the CALIOP track. A tiny hint of it (dark purple) is visible poking out from behind the CALIOP track at approximately 175 W, 62N, but it is really small and is best seen in the profile data. This is really a minor point that we could have omitted, but thought it best to describe for completeness.>

L353 'If the background spectrum has multiple modes (for example, one mode representing deep convective cloud radiances and another for cloud-free radiances), then multiple populations of the Monte-Carlo height samples may accumulate, leading to a multimodal height PDF': This should be discussed in the section about systematic error sources (see comment to L235).

**<The revision includes a much better discussion of this.>**

L374-390: In my opinion, the discussion about the e-folding-times is out of scope of the actual paper. Therefore, it would be better to skip it. (E.g. the authors do not discuss, that this is an apparent time because the effect of dilution and therefore not being detectable any more for the nadir-sounder is also included in this measure.)

<We have made it clear in the revision that this is an "apparent" e-folding time encompassing all sources of SO2 loss including chemistry as well as dilution and detection threshold-induced loss. We feel that this is a valuable analysis, especially after several weeks since it is (to our knowledge) the first of its kind for the Raikoke eruption cloud and that the ability to generate probabilistic time series are the logical extension of the uncertainty propagation at the core of this work.> L391 'Conclusions':Please discuss also the limitations of the method applied to CrIS: no retrievals in nue\_1 band of SO2, only linear assumption for retrieval of total column amounts.

<The revised conclusions are more comprehensive with respect to these issues.>

L402 'improved spatial resolution over IASI': Please mention here explicitly the CrIS and IASI pixel-size (km x km). What is the S/N in comparison of the two instruments?

<We have included comparisons of the two instruments.>

L414 'analysis of errors in the trace gas technique induced by a warming background atmosphere': What does this mean? Please provide an example.

<We have omitted this as it outside our scope.>

Technical comments: L24 'subtly':-> 'subtle.

L27 'Because': -> 'Because'

L308 '(f)':-> '(d)' (first appearance)

L395 'exceedence':-> 'exceedance'

Fig. 11:The difference between 'red' and 'blue', and (a) and (b) are not clear.

L412 'applyication':-> 'application'

L427 'CDF': Please spell out.

<We have fixed these.>

References:Carboni, E., Grainger, R., Walker, J., Dudhia, A., and Siddans, R.: A new scheme for sulphur dioxide retrieval from IASI measurements: Application to the Eyjafjalla-jökull eruption of April and May 2010, Atmos. Chem. Phys., 12, 11417–11434,doi:10.5194/acp-12-11417-2012, 2012.

Clarisse, L., Coheur, P.-F., Theys, N., Hurtmans, D., and Clerbaux, C.: The 2011 Nabro eruption, a SO2 plume height analysis using IASI measurements, Atmos. Chem.Phys., 14, 3095–3111, doi:10.5194/acp-14-3095-2014, 2014.

---

## Author Response (AR2)

Final Author Response:

Probabilistic retrieval of volcanic SO2 layer height and partial column density using the Cross-track Infrared Sounder (CrIS)

AMT-2020-41
* * *
* * *
* * *
* * *
We have made all of the listed technical corrections, except that in the first (L47: measurments -> measurements'), we have made the spelling correction, but not included the apostrophe as there is no possessive form implied in the parenthetical clause of this sentence.